# Benchmarking Multimodal AutoML
# for Tabular Data with Text Fields

**Xingjian Shi**[*]          xjshi@amazon.com
**Jonas Mueller**[*]         jonasmue@amazon.com
**Nick Erickson**            neerick@amazon.com
**Mu Li**                    mli@amazon.com
**Alexander J. Smola**       alex@smola.org

Amazon Web Services

## Abstract

We consider the use of automated supervised learning systems for data tables that not only contain numeric/categorical columns, but one or more text fields as well. Here we assemble 18 multimodal data tables that each contain some text fields and stem from a real business application. Our publicly-available benchmark[2] enables researchers to comprehensively evaluate their own methods for supervised learning with numeric, categorical, and text features. To ensure that any single modeling strategy which performs well over all 18 datasets will serve as a practical foundation for multimodal text/tabular AutoML, the diverse datasets in our benchmark vary greatly in: sample size, problem types (a mix of classification and regression tasks), number of features (with the number of text columns ranging from 1 to 28 between datasets), as well as how the predictive signal is decomposed between text vs. numeric/categorical features (and predictive interactions thereof). Over this benchmark, we evaluate various straightforward pipelines to model such data, including standard two-stage approaches where NLP is used to featurize the text such that AutoML for tabular data can then be applied. Compared with human data science teams, the fully automated methodology[3] that performed best on our benchmark also manages to rank 1st place when fit to the raw text/tabular data in two MachineHack prediction competitions and 2nd place (out of 2380 teams) in Kaggle's Mercari Price Suggestion Challenge.

## 1  Introduction

Despite recent data proliferation, the practical value of machine learning (ML) remains hampered by an inability to quickly translate raw data into accurate predictions. Automatic Machine Learning (AutoML) aims to address this via pipelines that can ingest raw data, train models, and output accurate predictions, all without human intervention [35]. Given their immense potential, many AutoML systems exist for data structured in tables, which are ubiquitous across science/industry [24, 30, 56].

Many data tables contain not only numeric and categorical fields (together referred to as *tabular* here), but also fields with free-form text. For example, Table 1 depicts actual data from the website Kickstarter. These contain multiple text fields such as the title and description of each funding proposal, numerical fields like the goal amount of funding and when the proposal was created, as well as categorical fields like the funding currency or country. This paper considers tables of this form where rows contain IID training examples (each with a single numeric/categorical value to predict,

---

[*]Equal contribution.
[2]Benchmark is available at: `https://github.com/sxjscience/automl_multimodal_benchmark`
[3]Open-source available to easily run on your own data: `https://github.com/awslabs/autogluon`

35th Conference on Neural Information Processing Systems (NeurIPS 2021) Track on Datasets and Benchmarks.

| name | desc | goal | country | currency | created_at | final_status |
|---|---|---|---|---|---|---|
| The Secret Order - The Game that gives back Gl... | Can you trust your friends? Solve the puzzle? ... | 5000.0 | GB | GBP | 1424101105 | 0 |
| Booker Family Foods. Home made, the way food s... | Community based, home-made-foods producer, to ... | 2500.0 | US | USD | 1404617242 | 0 |
| J.A.E.S.A : Next Generation Artificial Intelli... | A true next generation AI with the ability to ... | 30000.0 | CA | CAD | 1399078600 | 1 |

Table 1: Example of data in our multimodal benchmark with text (*name*, *desc*), numeric (*goal*, *created_at*), and categorical (*country*, *currency*) columns. From these features, we want to predict if a Kickstarter project will reach its funding goal or not (*final_status*).

i.e. regression/classification) and the columns used as predictive *features* can contain text, numeric, or categorical values. We refer to the value in a particular row and column as a *field*, where a single text field may actually contain a long text passage (e.g. a multi-paragraph item description). Despite their potential commercial value, there are currently few (automated) solutions for machine learning with this sort of data that jointly contain numeric/categorical and text features, which we here refer to as *multimodal* or *text/tabular* data. Applying existing AutoML tools to such data requires either manually featurizing text fields into tabular format [6, 28], or ignoring the text. Alternatively, one can model just the text with existing natural language processing (NLP) tools. [12, 26, 27, 34, 50].

This paper provides foundational tools aiming to spur a practical line of research that evaluates fundamental design choices for automated supervised learning with multimodal datasets that jointly contain text, numeric, and categorical features. Even though text commonly appears along with numeric/categorical fields in enterprise data tables, how to model such multimodal data has not been well studied in the literature. This stems from a lack of public benchmarks, as well as existing beliefs that basic featurization of the text should suffice for tabular models to exhibit strong performance [15, 28]. Here we introduce a new benchmark of 18 multimodal text/tabular datasets involving regression/classification tasks related to real business applications (Section 3), and provide a first systematic evaluation of some generic strategies for supervised learning with such data (Section 5).

Note that we write *AutoML* to describe any single modeling strategy that remains robustly performant across a diverse set of datasets without manual adjustments. The construction of an effective AutoML system critically relies on having an empirical benchmark of diverse datasets that are representative of real applications the system will be subsequently used for (in order to ensure the system performs well on the right types of data and not only on certain limited types of data). The experiments over our benchmark presented in this paper merely entail a preliminary evaluation of various straightforward (automated) multimodal modeling strategies that today's data scientists might consider for supervised learning with text/tabular data. Among other discoveries, our benchmark reveals that the conventional strategy of neural embeddings to featurize text for tabular models can be outperformed by simple alternatives. The strategy found to perform best in our benchmark (stack ensembling of tabular models with a multimodal Transformer network) should serve as a strong foundation for multimodal text/tabular AutoML[4], whose efficacy was subsequently verified in a few data science competitions (demonstrating that our benchmark and analysis have led to important new insights). That said, much further research is needed in this area, and we hope the public benchmark and open-source tooling introduced here will facilitate practical advances in important text/tabular modeling applications.

## 2 Related Work

Many ML courses teach data as vectors in $\mathbb{R}^d$, which is not the case in many practical applications. Thanks to the ubiquity of relevant benchmark data, substantial research has been conducted for properly handling categorical features in a unified manner that generalizes across datasets without sacrificing accuracy [25, 41, 47]. Given the prevalence of tabular data composed of numeric/categorical features, automating the ML process for such data has been the subject of extensive inquiry with major practical impact [24, 30, 35]. We hope our benchmark spurs similar progress on how to effectively handle text features in tabular data.

---

[4]A tutorial to easily run this method on your own text/tabular data is provided at: `https://auto.gluon.ai/stable/tutorials/tabular_prediction/tabular-multimodal-text-others.html`

Today, tools for automated learning with text data remain scarce (e.g. this dearth forced Blohm et al. [6] to turn to tabular AutoML tools for automated text prediction). Instead modern NLP applications primarily require experts who mostly favor Transformer networks as their model of choice for text [14, 48, 50]. However existing methods to input numeric/categorical features into Transformers remain rudimentary [50] and fail to outperform the best tree models for tabular prediction [33]. While multimodal text/tabular Transformer models have been utilized for *table understanding* tasks such as: semantic parsing of facts, cell filling, or relation extraction [13, 67], how to best adapt these models for standard classification/regression tasks with text/tabular features remains unstudied to our knowledge (a question that our benchmark might help answer). The use of tabular models together with Transformer-like text architectures has received limited attention [39, 62], and it remains unclear how to optimally leverage their complementary strengths for multimodal data (due to lack of benchmarks). In contrast, a number of entirely-neural architectures have been proposed for multimodal settings [36, 51, 52, 65]. However the vast majority of these are for {image, text} data [3, 49, 53, 54], but the gap between neural networks and alternative models is far greater for images than for tabular data [33]. In short, it remains unclear what are the best generic ML pipelines for text/tabular data based on models available today.

Large, sufficiently diverse/representative, public benchmarks have spurred significant progress in tabular AutoML [16, 17, 24, 69] and NLP [22, 40, 46, 63]. However we are not aware of any analogous benchmarks for evaluating multimodal text/tabular ML. There do exist a few miscellaneous text/tabular datasets scattered throughout popular ML data repositories [2, 59], but these are mostly small academic datasets that are not representative of modern applications with significant practical value. In contrast, multiple prediction competitions each involving a single real-world text/tabular dataset have been held, but winning solutions have heavily relied on dataset/domain-specific tricks of limited generalizability (c.f. mercari [45] and jigsaw dataset descriptions in Section 3.1). Here we aggregate multimodal datasets from competitions and other industry sources into one benchmark that aims to reveal unifying principles for powerful generic modeling of this form of data.

## 3 A Benchmark for Supervised Learning with Text/Tabular Data

In designing practical systems for real-world data tables that often contain text, the empirical performance of our design decisions is what ultimately matters. Representative benchmarks comprised of many diverse datasets are critical for proper evaluation of AutoML, whose aim is to reliably produce reasonable accuracy on arbitrary datasets without manual user-tweaking. Thus we introduce the first public benchmark for evaluating multimodal text/tabular ML, which is comprised of 18 tabular datasets, each containing at least one text field in addition to numeric/categorical columns. Our new benchmark is publicly available, as is the code to reproduce all results presented here.

Our benchmark strives to represent the types of ML tasks that commonly arise in industry today. In creating the benchmark, we aimed to include a mix of classification vs. regression tasks and datasets from real applications (as opposed to toy academic settings) that contain a rich mix of text, numeric, and categorical columns. Table 2 shows it is comprised of datasets that are quite

| Dataset ID | #Train | #Test | #Cat. | #Num. | #Text | Task | Metric | Prediction Target |
|---|---|---|---|---|---|---|---|---|
| prod | 5,091 | 1,273 | 1 | 0 | 1 | multiclass | accuracy | sentiment associated with product review |
| salary | 15,841 | 3,961 | 1 | 0 | 5 | multiclass | accuracy | salary range in data scientist job listings |
| airbnb | 18,316 | 4,579 | 37 | 24 | 28 | multiclass | accuracy | price label of Airbnb listing |
| channel | 20,284 | 5,071 | 1 | 15 | 1 | multiclass | accuracy | news category to which article belongs |
| wine | 84,123 | 21,031 | 0 | 2 | 3 | multiclass | accuracy | which variety of wine (type of grape) |
| imdb | 800 | 200 | 0 | 7 | 4 | binary | roc-auc | whether film is a drama |
| fake | 12,725 | 3,182 | 2 | 0 | 3 | binary | roc-auc | whether job postings are fake |
| kick | 86,502 | 21,626 | 3 | 3 | 3 | binary | roc-auc | whether proposed Kickstarter project will achieve funding goal |
| jigsaw | 100,000 | 25,000 | 2 | 27 | 1 | binary | roc-auc | whether social media comments are toxic |
| qaa | 4,863 | 1,216 | 1 | 0 | 3 | regression | $R^2$ | subjective type of answer (in relation to question) |
| qaq | 4,863 | 1,216 | 1 | 0 | 3 | regression | $R^2$ | subjective type of question (in relation to answer) |
| book | 4,989 | 1,248 | 1 | 2 | 5 | regression | $R^2$ | price of books |
| jc | 10,860 | 2,715 | 0 | 2 | 3 | regression | $R^2$ | price of JC Penney products on their website |
| cloth | 18,788 | 4,698 | 2 | 1 | 3 | regression | $R^2$ | customer review score for clothing item |
| ae | 22,662 | 5,666 | 3 | 2 | 6 | regression | $R^2$ | price of American-Eagle inner-wear items on their website |
| pop | 24,007 | 6,002 | 1 | 2 | 1 | regression | $R^2$ | online popularity of news article |
| house | 37,951 | 9,488 | 1 | 18 | 20 | regression | $R^2$ | sale price of houses in California |
| mercari | 100,000 | 25,000 | 3 | 0 | 6 | regression | $R^2$ | price of Mercari online marketplace products |

Table 2: The 18 multimodal datasets that comprise our benchmark. '#Cat.', '#Num.' and '#Text' count the number of categorical, numeric, and text features in each dataset, and '#Train' (or '#Test') count the training (or test) examples. In PDF, click on each Dataset ID for link to original data source.

diverse in terms of: sample-size, problem types, number of features, and type of features. 11 of the datasets contain more than one text field (with 28 text fields in the airbnb dataset). These text fields greatly vary in the amount of text they contain (e.g. short product names vs. lengthy product descriptions/reviews). The data (and text vocabulary) stem from a mix of of real-world domains spanning: e-commerce, news, social media, question-answering, and product listings (jobs, projects, films, Airbnb). Subsequent accuracy results from Table 3 indicate the 18 prediction problems also vary greatly in terms of both difficulty and how the predictive signal is divided between text/tabular modalities. To reflect real-world ML issues, we processed the data minimally (beyond ensuring the features/labels correspond to meaningful prediction tasks without duplicate examples) and thus there are arbitrarily-formatted strings and missing values all throughout. Methods/systems that perform well across the diverse set of 18 benchmark datasets are thus likely to provide real-world value for an important class of applications.

Each dataset in our benchmark is provided with a prespecified training/test split (usually 20% of the original data reserved for test set). Methods are not allowed to access the test set during training, and for validation (model-selection, hyperparameter-tuning, etc.) instead must themselves hold-out some data from the provided training data. As the choice of training/validation split is a key design decision in AutoML, we leave this flexible for different systems to choose in the learning process. To facilitate comparison between the different AutoML pipelines presented in this paper, we always used the same AutoGluon-provided training/validation split, which is stratified based on labels in classification tasks. Our use of other AutoML frameworks beyond AutoGluon (e.g. H2O) allows each framework to choose their own data splitting scheme.

The benchmark GitHub repository contains: (i) methods to easily retrieve the individual datasets and train/test splits, (ii) code to run all of the ML strategies studied in this paper and reproduce our results, and (iii) the scripts we used to produce each benchmark dataset from the original data source. Common modifications made to original data sources to produce the benchmark dataset versions included: defining a practically meaningful prediction task if there was not one associated with the original dataset, omitting duplicated rows, omitting non-predictive features (e.g. user ID) and those that were too correlated with the prediction target (making the benchmark too easy otherwise), applying log-transform to prediction targets that correspond to product prices (log-scale errors are more meaningful in most real pricing applications), and down-sampling overly large datasets (mercari, jigsaw) to ensure the benchmark remains computationally accessible.

### 3.1 Dataset Details

Each dataset can be easily loaded into Python (or another programming language) as the standard dataframe format used by `pandas`. All tables in our benchmark are appropriately formatted for supervised learning, with the first row serving as a standard header whose columns specify the names of each feature. We release our modified versions of the datasets in our benchmark under a **CC BY-NC-SA** license, and note that any data from this benchmark which has previously been published elsewhere falls under the original license from which the data originated (links to the original sources are provided). We the authors bear all responsibility in case of violation of rights. Long-term preservation of our benchmark is ensured by hosting the repository on GitHub, such that users can contribute their own improvements or publicly raise issues for us to address. The data files are hosted in AWS Simple Cloud Storage (S3), a reliable medium that will ensure researchers can easily obtain these files. Appendix D provides a datasheet [80] for our overall benchmark.

**prod**: Classify the sentiment (4-way classification) of user reviews of products based on the review text and product type (e.g. Tablet, Mobile, etc.). Intuitively, we expect most of the predictive signal to lie in the text, but predictions can be further improved by accounting for the fact that certain types of products tend to receive certain user sentiment. Representing a relatively simple multimodal task with only a single text feature and one categorical feature, this dataset originally stems from a 2020 MachineHack prediction competition: `https://machinehack.com/hackathons/product_sentiment_classification_weekend_hackathon_19/overview`

**salary**: Predict the salary range listed in data scientist job postings (in India) given the job description as well as other features like skill requirements and location. Intuitively, the best models will learn to identify valuable requirements from the text and high salary locations (via categorical modeling) as well as predictive interaction-effects. Representing a task with many text fields, this dataset originally stems from a 2018 MachineHack prediction competition: `https://machinehack.com/`

```
hackathons/predict_the_data_scientists_salary_in_india_hackathon/overview
```

**airbnb**: Predict the price label of AirBnb listings (in Melbourne, Australia) based on information from the listing page including various text descriptions and many numeric features (e.g. host's response-rate, number of bed/bath-rooms) and categorical features (e.g. property type, superhost or not). Representing a complex classification task with many features from each modality, the original version of this dataset was released via the InsideAirbnb initiative: `https://www.kaggle.com/tylerx/melbourne-airbnb-open-data`

**channel**: Predict which news category (i.e. channel) a Mashable.com news article belongs to based on the text of its title, as well as auxiliary numerical features like the number of words in the article, its average token length, how many keywords are listed, etc. Representing a task with one text field but many tabular (numeric) features, the original version of this dataset was collected by [20]: `https://archive.ics.uci.edu/ml/datasets/online+news+popularity`

**wine**: Classify the variety of wines based on tasting descriptions from sommeliers, and numeric features like price and categorical features like country-of-origin. The original version of this dataset was collected from WineEnthusiast: `https://www.kaggle.com/zynicide/wine-reviews`

**imdb**: Predict whether or not a movie falls within the Drama category based on text features like its name, description, actors/directors, and numerical features like its release year, runtime, etc. Representing a task with smaller sample-size, the original version of this dataset was collected from IMDB (the most popular movies): `https://www.kaggle.com/PromptCloudHQ/imdb-data`*

**fake**: Predict whether online job postings are real or fake based on their text and additional tabular features like amount of salary offered and degree of education required. Representing an imbalanced binary classification task, these data stem from the Employment Scam Aegean Dataset collected by [61]: `https://www.kaggle.com/shivamb/real-or-fake-fake-jobposting-prediction`

**kick**: Predict whether a proposed Kickstarter project will achieve funding goal based on text features like its title, description, numeric features like the amount of money requested, date posted, and categorical features like the country, currency, etc. This dataset represents a complex task where models must consider interactions between modalities to address a core question of Kickstarter's business: `https://www.kaggle.com/codename007/funding-successful-projects`

**jigsaw**: Predict whether online social media comments are toxic based on their text and additional tabular features providing information about the post (e.g. likes, rating, date created, etc.). This dataset originates from a 2019 Kaggle competition (`https://www.kaggle.com/c/jigsaw-unintended-bias-in-toxicity-classification`) in which the 1st place solution[5] utilized dataset-specific tricks such as a Bucket Sequencing Collator, auxiliary domain-specific prediction tasks for models, and a custom mimic loss function for training.

**qaa**: Given a question and an answer (from the Crowdsource team at Google) as well as an additional category feature, predict the (subjective) type of the answer in relation to the question. Representing a predominantly NLP task that requires deep language understanding (though the most accurate models must also consider the category), this dataset stems from a 2019 Kaggle competition: `https://www.kaggle.com/c/google-quest-challenge`

**qaq**: Given a question and an answer (from the Crowdsource team at Google) as well as additional category features, predict the (subjective) type of the question in relation to the answer. These data stem from the same source as **qaa**, where the different labels were both prediction targets in the original (multi-label) Kaggle competition.

**book**: Predict the sale price of books based on text features like their title, author, synopsis, categorical features like genre and numeric features like customer reviews and overall rating. This dataset originally stems from a 2019 MachineHack prediction competition: `https://machinehack.com/hackathons/predict_the_price_of_books/overview`

**jc**: Predict the sale price of items sold on the website of the retailer JC Penney based on text features like its title/description, and numeric features like its rating. Representing an important (e)commerce

---

*PromptCloud released the original version of the data from which we created this benchmark dataset.

[5] `https://www.kaggle.com/c/jigsaw-unintended-bias-in-toxicity-classification/discussion/103280`

task, this data was originally collected using information from the online page for each product: `https://www.kaggle.com/PromptCloudHQ/all-jc-penny-products*`

**cloth**: Predict the score of a customer review of clothing items (sold by an anonymous retailer) based on the review text, how much positive feedback the review has received (numeric), and additional features about the product like its department (categorical). The data were collected by [1]: `https://www.kaggle.com/nicapotato/womens-ecommerce-clothing-reviews`

**ae**: Predict the price of inner-wear items sold by retailer American Eagle based on text features like their product name, description, categorical features like brand, and numeric features like rating, review count. Representing an important (e)commerce task, this data was originally collected using information from the online page for each product: `https://www.kaggle.com/PromptCloudHQ/innerwear-data-from-victorias-secret-and-others*`

**pop**: Predict the popularity (number of shares on social media, on log-scale) of Mashable.com news articles based on the text of their title, as well as auxiliary numerical features like the number of words in the article, its average token length, and how many keywords are listed, etc. This dataset represents a very difficult prediction problem with only weak signal offered by the observed features. It is fundamentally hard to forecast how popular an article will be based only on its title and crude numerical summary statistics. To be comprehensive, an AutoML benchmark should contain at least one challenging problem like this. While **pop** stems from the same original data source as **channel**, the two have different labels to predict and do not share exactly the same set of features.

**house**: Predict sale prices of California homes sold in 2020 based on a text summary written by the seller and various tabular features (e.g. bedroom number, home type, location, year built, parking). Representing a regression task with many features that are text and numeric, this dataset originally stems from a 2021 Kaggle prediction competition: `https://www.kaggle.com/c/california-house-prices`

**mercari**: Predict the price of items sold in the online marketplace of Mercari based on information from the product page like name, description, free shipping availability, etc. This data originates from a 2017 Kaggle competition (`https://www.kaggle.com/c/mercari-price-suggestion-challenge/`), in which 1st place[6] and 3rd place[7] engineered dataset-specific text features such as customized bag-of-words and character N-grams, carefully tuned learning-rate/batch-size schedules, and specially ensembled models in a dataset-specific manner.

## 4 Text/Tabular Modeling Pipelines

Using our benchmark, we conduct a systematic empirical analysis of various baseline strategies for modeling text/tabular data. The strategy that performs best across the benchmark can serve as a promising starting point for automated supervised learning with multimodal data tables that contain text. Key choices include what models to use (and for which features), and how to optimally combine different models within an overall supervised learning pipeline. Our study considers popular modeling paradigms used by practitioners today, including: NLP models to featurize text for tabular models [6, 15, 28], ensembling of independently-trained text and tabular models [45], or end-to-end learning with neural networks that jointly operate on inputs across text and tabular modalities [36, 50, 51]. Below we merely outline the candidate strategies, Appendix A provides full descriptions.

**(Multimodal) Transformer Networks** Given their dominance across NLP, the only models we consider for handling raw text are popular Transformer neural networks which have initially been pretrained in an unsupervised fashion over a massive text corpus [11, 14, 44, 50, 60, 63]. We investigate how the end-to-end deep learning paradigm can be leveraged for simultaneous text and tabular inputs by extending standard Transformer networks into *multimodal Transformer networks* that jointly operate on both text and tabular features. Three multimodal network variants depicted in Figure S2 are considered: (1) *All-Text* – in which all tabular features are converted to strings and input into the Transformer as text, (2) *Fuse-Early* – in which dense embedding layers map the tabular features into the same vector space as the embedded text tokens such that self-attention and other

---

[6]`https://www.kaggle.com/c/mercari-price-suggestion-challenge/discussion/50256`
[7]`https://www.kaggle.com/c/mercari-price-suggestion-challenge/discussion/50272`

Transformer layers can be applied to learn low-level interactions across modalities, (3) *Fuse-Late* – a multi-tower network where one branch is a Transformer network for the text, other branches are multilayer perceptrons (MLP) for the numeric/categorical inputs, and the higher-level vector representations of each branch are pooled into a single multimodal vector representation (near the output layer of the overall network) via concatenation.

**Combining Transformers and Tabular Models**   As shown in Figure S3a, we consider featurizing the text into vector format followed by subsequent application of various tabular models [6]. Here the text embedding may stem from a pretrained Transformer network that has not been fine-tuned on our data (*Pre-Embedding*), a Transformer only trained on our text fields alone (*Text-Embedding*), or a multimodal Transformer network trained on both our text and tabular fields (*Multimodal-Embedding*). Note that 'tabular models' throughout are those trained on only numeric/categorical features, e.g. types of tree-based models. We also consider simple weighted ensembles that linearly combine the predictions of our Transformer network and various tabular models (*Weighted-Ensemble*, shown in Figure S3b) where each model takes as input the modalities it is suited for and is independently trained from the other models [9, 16]. Finally, stack ensembling is alternatively considered to nonlinearly aggregate predictions from the Transformer and tabular models (*Stack-Ensemble*, shown in Figure S3c), where an additional tabular 'stacker' model is trained using as its features the predictions output by the independently-trained Transformer and original tabular models [16, 58].

In our study, all tabular (numeric/categorical) modeling is simply done via AutoGluon-Tabular, an easy to use open-source tool for automated supervised learning on tabular data [16]. We chose AutoGluon because it has been found to produce highly accurate models for diverse tabular datasets [5, 18, 19, 68]. AutoGluon trains and ensembles a diverse suite of popular models for tabular data, including: Gradient Boosted Decision Trees [10, 38, 47], Extremely Randomized Trees [23], and MLP Neural Networks [16]. While neural networks are typically favored for unstructured data like text, decision tree ensembles have proven to be one of the most consistently performant models for tabular data [4, 18, 33]. Thus an effective strategy for text/tabular AutoML likely needs to appropriately combine the complementary strengths of Transformers and (tree-based) tabular models.

## 5   Experiments

To keep our study tractable, we adopt a sequential decision making process that decomposes the overall supervised learning pipeline design into three stages: 1) determine the appropriate Transformer backbone and fine-tuning strategy for text data alone (Appendix A.1), 2) determine the best way to extend this Transformer to text and tabular inputs (Appendix A.2), and 3) determine the best method to combine the best text and tabular models (Appendix A.3 and A.4). At each subsequent stage of the study, we explore modeling choices that are specific to that stage and simply use the best choice found in the empirical comparisons of the options available in previous stages.

For straightforward comparison, we employ the most commonly used classification/regression evaluation metrics that lie in $[0, 1]$ for reasonable predictions, with higher values indicating superior performance. We evaluate regression tasks via the coefficient of determination $R^2$, multiclass classification tasks via accuracy, and binary classification tasks via area under the ROC curve (AUC).

**Choice of Transformer Backbone**   Our first decision concerns the Transformer network itself, including what architecture and pretraining objective to employ. Existing results may not translate to our setting, since Transformers are typically applied to datasets with at most a couple text fields per training example [63, 64]. Here we choose between the (standard, already pretrained) base version of RoBERTa [44] or ELECTRA [11], two popular backbones used across modern NLP applications.

We first fine-tune the pretrained Transformer models as our sole predictors, using only the text features in each dataset. This helps identify which model is better at handling the types of text in our multimodal datasets. During fine-tuning of both of the RoBERTa or ELECTRA networks, we additionally consider two tricks to boost performance: 1) Exponentially decay the learning rate of the network parameters based on their depth [55]. We use a per-layer learning rate multiplier of $\tau^d$ in which $d$ is the layer depth and $\tau$ is the decay factor (set $= 0.8$ throughout). 2) Average the weights of the models loaded from the top-3 training checkpoints with the best validation scores [60].

The first section of Table 3 shows that ELECTRA performs better than RoBERTa across the text columns in our benchmark datasets. Our exponential decay and checkpoint-averaging tricks further

| Method | prod | qaq | qaa | cloth | airbnb | ae | mercari | jigsaw | imdb | fake | kick | jc | wine | pop | channel | salary | book | house | avg ↑ | mrr ↑ |
|---|---|---|---|---|---|---|---|---|---|---|---|---|---|---|---|---|---|---|---|---|
| *Choosing Text-Net:* | | | | | | | NLP Backbones and Fine-tuning Tricks (Section A.1) | | | | | | | | | | | | | |
| RoBERTa | 0.588 | 0.412 | 0.268 | 0.700 | 0.344 | 0.953 | 0.561 | 0.960 | 0.731 | 0.929 | 0.751 | 0.615 | 0.811 | -0.000 | 0.301 | 0.396 | 0.151 | 0.821 | 0.572 | 0.07 |
| ELECTRA | 0.705 | 0.410 | 0.356 | 0.718 | 0.349 | 0.955 | 0.586 | 0.965 | 0.750 | 0.824 | 0.754 | 0.606 | 0.813 | 0.003 | 0.315 | 0.457 | 0.466 | 0.857 | 0.605 | 0.09 |
| + Exponential Decay $\tau = 0.8$ | 0.728 | 0.436 | 0.431 | 0.743 | 0.337 | 0.953 | 0.579 | 0.963 | 0.852 | 0.963 | 0.760 | 0.664 | 0.808 | 0.004 | 0.308 | 0.447 | 0.568 | 0.841 | 0.632 | 0.12 |
| + Average 3 ★ | 0.729 | 0.451 | 0.432 | 0.746 | 0.350 | 0.954 | 0.581 | 0.965 | 0.858 | 0.961 | 0.766 | 0.656 | 0.807 | 0.004 | 0.307 | 0.445 | 0.571 | 0.841 | 0.635 | 0.14 |
| *Choosing Multimodal-Net:* | | | | | | | Fusion Strategy (Section A.2, Figure S2) | | | | | | | | | | | | | |
| All-Text | 0.907 | 0.454 | 0.419 | 0.746 | 0.366 | 0.957 | 0.599 | **0.967** | 0.840 | 0.967 | **0.799** | 0.645 | 0.810 | 0.013 | 0.480 | 0.465 | 0.585 | 0.892 | 0.662 | 0.28 |
| Fuse-Early | **0.913** | 0.441 | 0.418 | 0.745 | 0.377 | 0.953 | 0.596 | **0.967** | 0.843 | 0.960 | 0.770 | 0.653 | 0.806 | 0.013 | 0.474 | 0.458 | 0.548 | 0.901 | 0.658 | 0.21 |
| Fuse-Late ★ | 0.907 | 0.449 | **0.445** | 0.747 | 0.395 | 0.958 | 0.603 | 0.966 | 0.857 | 0.961 | 0.773 | 0.639 | 0.812 | 0.015 | 0.481 | 0.468 | 0.571 | 0.907 | 0.664 | 0.22 |
| *Choosing Aggregation:* | | | | | | | Multimodal Model Aggregation (Sections A.3 and A.4, Figure S3) | | | | | | | | | | | | | |
| Pre-Embedding | 0.895 | 0.216 | 0.247 | 0.642 | 0.449 | 0.972 | 0.433 | 0.586 | 0.871 | 0.926 | 0.743 | 0.491 | 0.680 | 0.012 | 0.526 | 0.460 | 0.581 | 0.939 | 0.593 | 0.11 |
| Text-Embedding | 0.867 | 0.446 | 0.432 | 0.748 | 0.430 | 0.972 | 0.434 | 0.587 | 0.855 | 0.962 | 0.790 | 0.658 | 0.830 | 0.008 | 0.502 | 0.438 | 0.594 | 0.932 | 0.638 | 0.17 |
| Multimodal-Embedding | 0.907 | 0.439 | 0.437 | 0.749 | 0.438 | 0.974 | 0.432 | 0.587 | 0.847 | 0.967 | 0.794 | **0.683** | 0.829 | 0.007 | 0.517 | 0.451 | 0.595 | 0.934 | 0.644 | 0.25 |
| Weighted-Ensemble | 0.907 | 0.439 | 0.429 | 0.744 | 0.453 | 0.976 | 0.597 | 0.957 | 0.876 | 0.923 | 0.787 | 0.641 | 0.814 | 0.018 | 0.554 | 0.483 | 0.620 | 0.941 | 0.676 | 0.25 |
| Stack-Ensemble ★ | 0.909 | **0.456** | 0.438 | **0.751** | 0.459 | 0.977 | **0.605** | **0.967** | **0.878** | 0.964 | 0.797 | 0.624 | 0.836 | **0.020** | **0.556** | 0.496 | **0.638** | **0.943** | **0.684** | **0.69** |
| | | | | | | | Tabular AutoML + Feature Engineering Baselines (Section A.3) | | | | | | | | | | | | | |
| AG-Weighted | 0.891 | 0.046 | 0.076 | -0.002 | 0.426 | 0.841 | 0.098 | 0.587 | 0.845 | 0.686 | 0.668 | 0.004 | 0.173 | 0.016 | 0.549 | 0.226 | 0.222 | 0.934 | 0.405 | 0.08 |
| AG-Stack | 0.891 | 0.046 | 0.077 | 0.001 | 0.435 | 0.841 | 0.098 | 0.587 | 0.844 | 0.697 | 0.670 | 0.003 | 0.175 | 0.017 | 0.550 | 0.226 | 0.233 | 0.934 | 0.407 | 0.09 |
| AG-Weighted+ N-Gram | 0.892 | 0.426 | 0.382 | 0.610 | 0.450 | 0.978 | 0.526 | 0.909 | 0.842 | 0.966 | 0.772 | 0.357 | 0.829 | 0.019 | 0.546 | 0.484 | 0.591 | 0.941 | 0.640 | 0.17 |
| AG-Stack+ N-Gram | 0.895 | 0.414 | 0.383 | 0.654 | **0.466** | **0.979** | 0.569 | 0.915 | 0.850 | **0.968** | 0.775 | 0.612 | **0.842** | **0.020** | 0.548 | 0.494 | 0.600 | 0.943 | 0.663 | 0.43 |
| H2O AutoML | 0.869 | 0.247 | 0.159 | 0.163 | 0.329 | 0.976 | 0.430 | 0.531 | 0.813 | 0.756 | 0.669 | 0.411 | 0.478 | 0.014 | 0.530 | 0.525 | 0.444 | 0.939 | 0.516 | 0.11 |
| H2O AutoML + Word2Vec | 0.859 | 0.244 | 0.285 | 0.624 | 0.347 | 0.973 | 0.534 | 0.847 | 0.827 | 0.943 | 0.755 | 0.443 | 0.778 | 0.013 | 0.524 | **0.528** | 0.586 | 0.932 | 0.613 | 0.14 |
| H2O AutoML + Pre-Embedding | 0.846 | 0.227 | 0.312 | 0.644 | 0.367 | 0.969 | 0.282 | 0.572 | 0.874 | 0.893 | 0.738 | 0.549 | 0.571 | 0.007 | 0.483 | 0.483 | 0.523 | 0.933 | 0.572 | 0.09 |

Table 3: Accuracy (and $R^2$, AUC) of AutoML strategies over our multimodal benchmark. Column **avg** lists each method's average score across datasets (i.e. how *much* methods differ in overall performance) and **mrr** its mean reciprocal rank among all evaluated methods (i.e. how *often* a method outperforms others). Each subsection encapsulates a design stage (★ marks variant with best avg).

boost performance, with the majority of additional gains produced by exponential decay. In subsequent experiments, we thus fix ELECTRA fine-tuned with both exponential decay and checkpoint-averaging as the model used to handle text features and call it *Text-Net*.

**Best Multimodal Network**    Next, we explore the best way to extend the *Text-Net* model to operate across numeric/categorical inputs in addition to text fields (among the options in Figure S2). Across our datasets, Table 3 shows that the *Fuse-Late* strategy outperforms *Text-Net* and the alternative *All-Text*/*Fuse-Early* options for producing predictions from multimodal inputs using a single neural network. We thus fix this *Fuse-Late* model as our *Multimodal-Net* used in subsequent experiments.

**Aggregating Transformers and Tabular Models**    Having identified a good neural network architecture for multimodal text/tabular inputs, we now study combinations of such models with classical learning algorithms for tabular data (among the options in Figure S3). Where not specified, the tabular models are those trained by AutoGluon-Tabular (see Appendix B.4). The third section of Table 3 illustrates that *Stack-Ensemble* is overall the best aggregation strategy. Ensembling the predictions of *Multimodal-Net* and tabular models is better than instead using the Transformer for text embedding. As expected, *Text-Embedding* and *Multimodal-Embedding* outperform *Pre-Embedding*, demonstrating how domain-specific fine-tuning improves the quality of learned embeddings. *Multimodal-Embedding* performs better than *Text-Embedding* on some datasets and similarly across the rest, showing it can be beneficial to use text representations contextualized on numeric/categorical information.

**AutoGluon Baselines**    As many of our results use the tabular models in AutoGluon [16], we also compare different variants of AutoGluon-Tabular (without our *Multimodal-Net*) as baselines:

*AG-Weighted* / *AG-Stack*: We train AutoGluon with weighted / stack ensembling of its tabular models, here ignoring all text columns. Thus, baseline ML performance of tabular models without using any text fields can be established via the AG-Weighted/Stack numbers in Table 3 (without N-Gram).

*AG-Weighted + N-Gram* / *AG-Stack + N-Gram*: Similar to *AG-Weighted* / *AG-Stack*, except we first use AutoGluon's N-Gram featurization [15] to encode all text in tabular form.

The performance gap between AutoGluon-Tabular with and without N-Grams can reveal (an approximate lower bound for) how much extra predictive value is provided by the text features in each dataset. Inspecting these gaps, we find that, compared to the tabular features, text features contain most of the predictive signal in some datasets (qaq, qaa, cloth, mercari, jc), and far less signal in other datasets (prod, imdb, channel), again highlighting the diversity of our benchmark. Note that our proposed *Stack-Ensemble* performs relatively well across all types of datasets, regardless how the predictive signal is allocated between text and tabular features.

**H2O Baselines**    In addition to AutoGluon, we also run another popular open-source AutoML tool offered in H2O [29]. Since H2O AutoML is not designed for the text in our multimodal data tables, we try combining H2O's NLP tool [28] and tabular AutoML tool [42].

*H2O AutoML*: We run H2O AutoML directly on the original data of our benchmark. As a tabular AutoML framework, H2O AutoML is assumed to ignore text features, but H2O categorizes feature types differently than us and automatically treats some columns we consider to be text as categorical instead.

*H2O AutoML + Word2Vec*: We run H2O's word2vec algorithm to featurize text fields and then H2O AutoML on the featurized data, following their recommended procedure [28].

*H2O AutoML + Pre-Embedding*: We featurize each text field using embeddings from a pretrained ELECTRA Transformer, as in *Pre-Embedding*, followed by H2O AutoML on the featurized data table.

The last section of Table 3 shows that while these powerful AutoML ensemble predictors can outperform our individual neural network models (particularly for datasets with more predictive signal in the tabular features), our *Stack-Ensemble* and *Weighted-Ensemble* are superior overall.

Table 3 shows the accuracy for some datasets significantly improves when models utilize the text features rather than ignoring them. Predictive performance of *AG-stack* (baseline tabular model that ignores text) vs. *Stack-Ensemble* (our extension that leverages text) is 0.098 vs. 0.605 on mercari, 0.670 vs. 0.797 on kick, and 0.175 vs. 0.836 on wine. On these datasets, modeling the tabular features brings clear improvements over the text alone given the performance of *Text-Net* (our best text Transformer model that ignores tabular features) is only: 0.581 on mercari, 0.766 on kick, and 0.807 on wine. In certain applications, accuracy improvements of this magnitude may have significant commercial value, thus highlighting the benefits of multimodal modeling of text/tabular data.

**Performance in Real-world ML Competitions**    Some datasets in our multimodal benchmark originally stem from previous ML competitions. For these (and other recent competitions with text/tabular data), we fit the automated strategy that performed best in our benchmark (*Stack-Ensemble*) to the official competition dataset, without manual adjustment or data processing. We then submit its resulting predictions on the competition test data to be scored, which enables us to see how they fare against the manual efforts of human data science teams.

The *Stack-Ensemble* strategy achieves 1st place historical leaderboard rank in two MachineHack prediction competitions: *Product Sentiment Classification*[8] and *Predict the Data Scientists Salary in India*[9], and 2nd place in another: *Predict the Price of Books*[10]. This same strategy also achieves 2nd place on the historical leaderboard of two Kaggle competitions: *California House Prices*[11] and *Mercari Price Suggestion Challenge*[12], where the latter was a very popular Kaggle competition in which 2380 teams participated (with $100,000 prize offered to winner). These results show that a straightforward AutoML strategy identified via preliminary analysis of our benchmark is already competitive with data scientists on real-world text/tabular datasets that possess great commercial value. Extensive studies of the benchmark will presumably reveal even more effective strategies.

## 6    Discussion

Lacking public benchmarks, academic research on ML for multimodal text/tabular data has not matched industry demand to derive practical value from such data. This paper provides evidence that generic best practices for such data remain unclear today: we simply evaluated a few basic strategies on our benchmark and found a single automated strategy that turns out to outperform top human data scientists in numerous historical prediction competitions involving diverse text/tabular data. This strategy uses a stack ensemble (Appendix A.4) of tabular models trained on top of predictions from other tabular models and a *Multimodal-Net* (depicted in Figure S3c). The latter network is based on a *Fuse-Late* architecture (depicted in Figure S2c) with concatenation of text, numeric, and categorical representations (where text representations are produced via the ELECTRA Transformer backbone) and is trained via fine-tuning with exponential learning rate decay and checkpoint averaging.

---

[8]`https://www.machinehack.com/hackathons/product_sentiment_classification_weekend_hackathon_19/overview` ("Anonymous Submission ID 1556" entry)

[9]`https://machinehack.com/hackathons/predict_the_data_scientists_salary_in_india_hackathon/overview` ("Xingjian Shi" entry)

[10]`https://machinehack.com/hackathons/predict_the_price_of_books/overview`

[11]`https://www.kaggle.com/c/california-house-prices` ("sxjscience" entry)

[12]`https://github.com/sxjscience/automl_multimodal_benchmark/blob/main/competition_submissions/mercari_submission_screenshot.png`

Using our benchmark, we conducted a systematic evaluation of a few generic strategies for supervised learning with such multimodal text/tabular data. The strategy that performed best over our benchmark (stack ensemble) also exhibits highly competitive performance in numerous text/tabular prediction competitions. This highlights the utility of our benchmark in revealing performant modeling techniques, indicating that the benchmark is sufficiently diverse and representative of real-world text/tabular prediction tasks. Our benchmark analysis challenges certain conventional beliefs:

- Neural embedding of text followed by tabular modeling (*Pre/Text-Embedding*) [6, 28] is often outperformed by N-gram featurization (*AG-Stack + N-Gram*) or leveraging predictions from text neural networks (*Stack-Ensemble*) rather than their representations (embeddings). Given the success of pretrained Transformers across NLP, we are surprised to find both N-Grams and word2vec here provide superior text featurization than *Pre-Embedding*.

- In the architecture of multimodal networks for classification/regression, newer ideas to fuse modalities in early layers (i.e. *Fuse-Early/All-Text* Transformers with cross-modality attention [32, 50, 53]) are not necessarily superior to older multi-tower *Fuse-Late* architectures that fuse representations in higher layers closer to the output [3, 36, 51].

- An end-to-end multimodal neural network is surpassed by stack ensembling this *Multimodal-Net* with tabular models trained in separate stages rather than end-to-end (*Stack-Ensemble*).

Previously anticipated conclusions that are empirically validated by our benchmark include:

- Text featurization is better via fine-tuned networks (*Text-Embedding*) than pretrained ones (*Pre-Embedding*), and slightly better via a fine-tuned multimodal network (*Multimodal-Embedding*), whose text embeddings benefit from contextualization on the tabular features.

- Able to exploit predictive interactions between different modalities, stack ensembling outperforms simple weighted ensembling, yet it still facilitates modular system design.

A general observation across our benchmark is that stacking/fusing models later helps more than fusing low-level features. While this finding contrasts with other multimodal ML research [49, 53], we suspect the primary reason is that most of this other multimodal ML has predominantly has focused on particular matching tasks with mostly image+text data (e.g. image captioning). Such tasks require learning to correlate low-level features of one modality (e.g. specific words) with low-level features in the other modality (e.g. specific pixel regions), and thus early-fusion is a natural modeling strategy. However, our benchmark is composed of commercially relevant classification/regression tasks with text+tabular data, in which we typically predict an auxiliary variable based on the text/tabular features (e.g. the price of a product). For such tasks, we suspect correlating low-level features between modalities is unnecessary and it is more critical that models can adequately summarize/extract the relevant information from each modality before considering the interaction between them. Using our benchmark, future work may more formally investigate such hypotheses.

**Conclusion** Further analysis of our benchmark can reveal many more practical ML insights, including under which data conditions certain methods perform better than others. Future research should investigate different data preprocessing pipelines, which are known to play an important role in AutoML. Other important questions not considered in our preliminary study include *how to best*: Handle many long text fields? Perform multimodal feature selection? Apply feature engineering that combines synergistically with learned neural network representations? Allocate limited training/HPO time between cheaper tabular models and more expensive text neural networks? We hope our public benchmark spurs the AutoML community to broaden their methods' applicability to more data types.

**Limitations and Societal Impact** Since our benchmark only contains text in the English language and primarily from commercial domains, its conclusions will only hold for particular types of applications. To ensure similar advancements for text/tabular data with low-resource languages [31, 37, 40], we encourage the development of a similar benchmark with non-English text. We also caution that analysis of text fields may raise privacy concerns as such fields may expose arbitrary personal information [8, 21]. Since text fields may contain arbitrary information, they are also prone to introducing spurious correlations in training data that may harm accuracy during deployment [57] and may be undesirably coupled to protected attributes such as race, gender, or socioeconomic status [66]. Basing automated business decisions on customer-generated text could also be more susceptible to adversarial manipulation [43] than tabular features that customers cannot as easily control.

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
