# Appendix

## A    Descriptions of Text/Tabular Modeling Pipelines

Here we fully describe the various straightforward modeling strategies that we evaluate over our benchmark in order to identify performant baselines for automated supervised learning with multimodal data tables that contain text. Recall our study aims to cover popular variants of text/tabular modeling used in practice today, including: NLP models to featurize text for tabular models [6, 15, 28], ensembling of independently-trained text and tabular models [45], or end-to-end learning with neural networks that jointly operate on inputs across text and tabular modalities [36, 50, 51]. We first consider the latter paradigm of multimodal neural network models, which in subsequent sections are also considered for text featurization and ensembling with tabular models.

### A.1    Transformer Models for Text

We first consider solely inputting the text into our neural network and then discuss how to extend the network to additional numeric/categorical inputs in Section A.2. While many neural architectures have been proposed to model text, pretrained Transformer networks now dominate modern NLP. These models are first pretrained in an unsupervised manner on a massive text corpus before being fine-tuned over our (smaller) labeled dataset of interest [14, 50]. This allows our supervised learning to benefit from information gleaned from the external text corpus that would otherwise not be available in our limited labeled data. The Transformer also effectively aggregates information from various aspects of a training example, using a *self-attention* mechanism to contextualize its intermediate representations based on particularly informative features [60]. Since BERT [14] first demonstrated the power of Transformer pretraining via Masked Language Modeling (MLM), superior pretraining techniques have been developed. RoBERTa [44] dynamically generates masks and pretrains on a larger corpus for a longer time, employing the same MLM objective as BERT in which random tokens are masked for the Transformer to guess their original value. ELECTRA [11] is an alternative pretraining technique in which a simple generative model randomly replaces tokens and the Transformer must classify which tokens were replaced.

Given a dataset with multiple text columns, we feed the tokenized text from all columns jointly into our Transformer (with special [SEP] delimiter tokens between fields and a [CLS] prefix token appended at the start [14]), as detailed in the next paragraph. A single embedding vector for all text fields is obtained from the Transformer's representation at the [CLS] position after feeding the merged input into the network [14]. Similarly, just a single text field can be embedded via the Transformer's vector representation at the [CLS] position, after feeding only this field into the network.

**Handling Multiple Text Fields in the Transformer**    Given multiple text columns, we feed the tokenized text from all columns jointly into our Transformer, as illustrated in Figure S1. We follow the usual method to format text from multiple passages [14]: tokenized inputs from different text fields are merged with special [SEP] delimiter tokens between fields and a [CLS] prefix token is subsequently appended at the start of merged input. To further ensure that the network distinguishes boundaries between adjacent text fields, we alternate 0s and 1s as the segment IDs. Here segment IDs and the [SEP] token were previously used to demarcate boundaries between passages during pretraining [14]. After feeding the merged inputs into the Transformer, we can extract its intermediate representations at each position as token-level embeddings (each token has one embedding, which has been contextualized based on information from the other tokens).

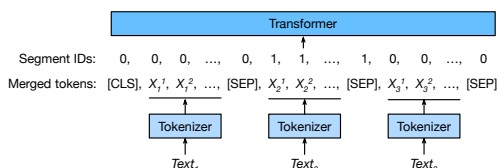

Figure S1: Inputting data from 3 text fields into Transformer.

When the total length of tokenized text fields exceed the maximum allowed length (set to be 512 throughout this work), we truncate the input by repeatedly removing one token from the longest

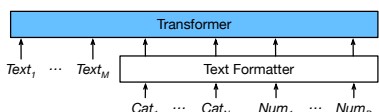

(a) *All-Text*. Convert numeric and categorical values into additional text tokens.

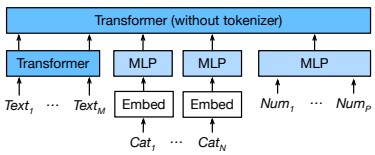

(b) *Fuse-Early*. Transformer operates on learned embeddings for each feature.

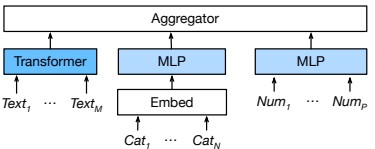

(c) *Fuse-Late*. Separate branches encode each modality, and aggregated via concatenation.

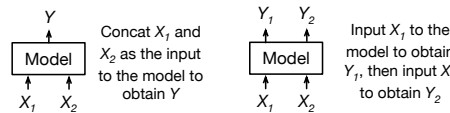

(d) Notation used in these figures.

Figure S2: Options for fusing modalities in *Multimodal-Net* (Section A.2). Two dense layers (not shown) are added on top of each network in (a)-(c) to output a prediction (real value for regression, logit vector for classification). Over our benchmark, option (c) performs best and is the chosen *Multimodal-Net* architecture that we subsequently try combining with tabular models.

individual text field until the length constraint is met. Since self-attention is permutation equivariant, a common practice is to assign an additional vector that encodes each position (namely positional encoding) so that the Transformer can distinguish between identical tokens occurring at different locations [60]. After merging multiple text fields into a single input, we simply assign positional encodings based on this larger input.

## A.2 Extending Transformer Architectures to Multimodal Inputs

In many multimodal datasets, some of the predictive signal solely resides in text fields, while other predictive information is restricted to tabular feature values, or complex interactions between text and tabular values. To enjoy the benefits of end-to-end learning without sacrificing accuracy, we consider how to adapt a Transformer network to simultaneously operate on inputs from both modalities, referring to the resulting network as *Multimodal-Net*. A natural approach in our setting is to enhance the Transformer such that its attention mechanism can contextualize representations of individual text tokens based not only on other parts of the text, but also on the values of relevant tabular features as well. Below we discuss three different options for implementing the *Multimodal-Net* that are depicted in Figure S2 (with details in Appendix B.2). These options differ in whether information is fused across text and tabular modalities: at the input layer (*All-Text*), in the earlier layers of the network near the input (*Fuse-Early*), or in the later layers of the network near the output (*Fuse-Late*).

**All-Text**    A simple (yet crude) option is to convert numeric and categorical values to strings and subsequently treat their columns also as text fields [50]. Through its byte-pair encoding, a pretrained Transformer can handle most categorical strings and may be able to crudely represent numeric values within a certain range (here we round all numbers to 3 significant digits in their string representation).

**Fuse-Early**    Rather than casting them as strings, we can allow our model to adaptively learn token representations for each numeric and categorical feature via backpropagation (see Figure S2b). We introduce an extra factorized embedding layer [25, 89] to map categorical values into the same $\mathbb{R}^d$ vector representation encoded by the pretrained Transformer backbone for text tokens (with different embedding layers used for different categorical columns in the table). All numeric features are encoded via a single-hidden-layer Multi-layer Perceptron (MLP) to obtain a unified $\mathbb{R}^d$ vector representation. The resulting $d$-dimensional vector representations from each modality are jointly fed into a 6-layer Transformer encoder whose self-attention operations can model interactions between the embeddings of text tokens, categorical values, and numeric values. We refer to this strategy as *Fuse-Early* because only a minimal (yet adaptive) input processing layer is added to convert the tabular features into a common vector form which can be jointly fed through many shared Transformer layers. Huang et al. [33] considered a similar strategy for applying Transformers to entirely numeric/categorical data, albeit without text components that are a major focus here.

**Fuse-Late**   Rather than aggregating information across modalities in early network layers, we can perform separate neural operations on each data type and only aggregate per-modality representations into a single representation near the output layer (see Figure S2c). This multi-branch design allows each branch to extract higher-level representations of the values from each modality, before the network needs to consider how modalities should be fused. Here we use a multi-tower architecture in which numeric and categorical features are fed into separate MLPs for each modality. The text features are fed into a (pretrained) Transformer network. The topmost vector representations of all three networks are pooled into a single vector from which predictions are output via two dense layers. As pooling operators, we considered mean/max pooling or concatenation as options. Experiments show these pooling methods perform similarly on each dataset, with concatenation exhibiting slightly better overall performance, and we thus fix concatenation as our pooling method in the Fuse-Late architecture.

### A.3   Featurizing Text for Tabular Models

Despite their success for modeling text, the application of Transformer architectures to tabular data remains limited [18, 33, 76]. The use of tabular models together with Transformer-like text architectures has also received little attention [39, 62]. Recall that 'tabular models' throughout are those trained on only numeric/categorical features, e.g. different types of decision tree ensembles fit by AutoGluon-Tabular.

To allow tabular models to access information in text fields, the text is typically first mapped to a continuous vector representation which replaces a text column in our data table with multiple numeric columns (one for each vector dimension). One can treat each text column as a document, and each individual text field as a paragraph within the document, such that each text field can be featurized via NLP methods for computing text representations [15, 52, 93] before the tabular models are trained.

Rather than classical NLP methods like N-grams or word embeddings [15], a Transformer can instead be used to map the text fields into a vector representation via contextual embedding [6, 14]. Subsequently, the text fields are replaced in the data table by additional numeric columns corresponding to each dimension of the embedding vector (Figure S3a). Our study considers three ways to featurize text using a Transformer.

**Pre-Embedding**   Most straightforward is to embed text via a pretrained Transformer (not fine-tuned on our labeled data), and subsequently train tabular models over the featurized data table [6].

**Text-Embedding**   The *Pre-Embedding* strategy is not informed about our particular prediction problem and the domain of the text data. In *Text-Embedding*, we further fine-tune the pretrained Transformer to predict our labels from only the text fields, and use the resulting Text-Net to embed the text. By adapting to the domain of the specific prediction task, *Text-Embedding* is able to extract more relevant textual features that can improve the performance of tabular models. This is particularly true in settings where the target only depends on one out of many text fields, since the fine-tuning process can produce representations that vary more based on the relevant field vs. irrelevant text.

**Multimodal-Embedding**   Text representations may improve when self-attention is informed by context regarding numeric/categorical features. Thus we also consider embedding text via our best multimodal network from Section A.2 (depicted in Figure S2c). These models are again fine-tuned using the labeled data and now produce a single vector representation for *all* columns in the dataset, regardless of their type. Since Transformers are better suited for modeling text than tabular features, we only replace the text fields with the learned vector, all other non-text features are kept and used for subsequent tabular learning. Thus the sole difference between *Text-Embedding* and *Multimodal-Embedding* is that the embeddings used to replace text are additionally contextualized on numeric/categorical feature values in the latter method.

### A.4   Aggregating Text & Tabular Models

Rather than merely leveraging the Transformers for their embedding vector representations as in Section A.3, an alternative multimodal text/tabular modeling strategy is to instead consider their predictions and ensemble these with predictions from tabular models. Utilized by most AutoML frameworks [16, 42, 78], model ensembling is a straightforward technique to boost predictive accuracy. Ensembling is particularly suited for multimodal data, where different models may be trained with

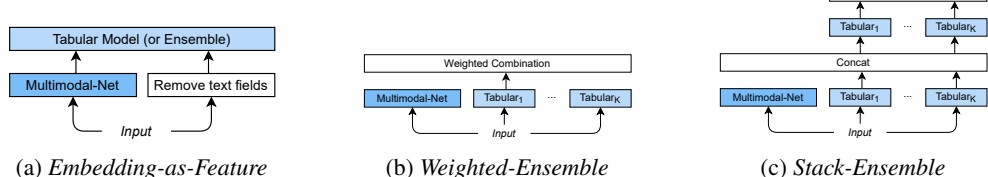

|  | (a) *Embedding-as-Feature* | (b) *Weighted-Ensemble* | (c) *Stack-Ensemble* |

Figure S3: Options for combining *Multimodal-Net* with classical tabular models. Five particular tabular models are used in this paper: extremely randomized trees, a simple MLP, and three different types of gradient boosted decision trees. Over our benchmark, option (c) performs the best and is chosen as the strategy for aggregating text and tabular models in our proposed AutoML solution.

different modalities. However, the resulting ensemble may then be unable to exploit nonlinear predictive interactions between features from different modalities. To remedy this, we advocate for the use of our multimodal Transformers (from Section A.2) that fuse information from text and tabular inputs. Here we specifically consider ensembling the multimodal Transformer model with the various standard tabular models used by AutoGluon-Tabular. Furthermore, we propose stack ensembling with nonlinear aggregation of model predictions that can exploit inter-modality interactions between different base models' predictions, even when base models do not overlap in modality.

**Weighted-Ensemble** We first consider straightforward aggregation via a weighted average of the predictions from our Transformer model and various tabular models (like those trained by AutoGluon-Tabular). Here, our Transformer and other models are independently trained using a common training/validation split. Subsequently, we apply *ensemble selection*, a forward-selection algorithm to fit aggregation weights over all models' predictions on the held-out validation data [9]. Unlike regression for fitting the aggregation weights [42, 58], ensemble selection is favored by many tabular AutoML tools like AutoGluon as it is more computationally efficient, less prone to overfitting, and naturally favors sparse weights [16, 79].

**Stack-Ensemble** Rather than restricting the aggregation to a linear combination, we can use stacking [103]. This trains another ML model to learn the best aggregation strategy. The features upon which the 'stacker' model operates are the predictions output by all base models (including our Transformer), concatenated with the original tabular features in the data. Following Erickson et al. [16], we try each type of tabular model in AutoGluon-Tabular as a stacker model (see Appendix B.4). To output predictions, a weighted ensemble is constructed via ensemble selection applied to the tabular stacker models (Figure S3c). We do not consider our larger (multimodal) Transformer model as a stacker since lightweight aggregation models are preferred in practice. Overfitting is a key peril in stacking, and we ensure that stacker models are only trained over *held out* predictions produced from base models via 5-fold cross-validation (bagging) [16, 58].

# B  Additional Experiment/Implementation Details

## B.1  Data Processing

For tabular features/models, we can simply rely on the same preprocessing as AutoGluon-Tabular, which has been found to also work well for other learning algorithms [16]. For our subsequently introduced multimodal neural networks that operate on both text and tabular features, we simply rescale and center numeric features and impute their missing values via their average. Missing values for categorical features (and previously unseen categories encountered during inference) are represented via an additional Unknown category in order to avoid unrealistic missing at random assumptions. Missing text fields are handled as empty strings in our preprocessing pipeline. The tabular MLP networks in AutoGluon-Tabular (the AutoML solution around which our experiments are based) also only use this simple preprocessing. We thus also utilized the same preprocessing for other neural networks evaluated in our experiments for controlled comparison. Note this was only done for the experiments presented here; the actual datasets in our benchmark have not been preprocessed in this manner, and the benchmark leaves preprocessing as a challenge for future AutoML systems to address as they see fit.

| Method | binary | multiclass | regression |
|---|---|---|---|
| RoBERTa | 0.843 | 0.511 | 0.501 |
| ELECTRA | 0.823 | 0.545 | 0.519 |
| + Exponential Decay $\tau = 0.8$ | 0.885 | 0.545 | 0.544 |
| + Average 3 ★ | 0.887 | 0.548 | 0.546 |
| *Choosing Multimodal-Net:* | | *Fusion Strategy* | |
| All-Text | 0.893 | 0.641 | 0.548 |
| Fuse-Early | 0.885 | 0.643 | 0.546 |
| Fuse-Late ★ | 0.889 | 0.649 | 0.551 |
| *Choosing Aggregation:* | | *Multimodal Model Aggregation* | |
| Pre-Embedding | 0.781 | 0.638 | 0.430 |
| Text-Embedding | 0.798 | 0.657 | 0.528 |
| Multimodal-Embedding | 0.799 | 0.673 | 0.532 |
| Weighted-Ensemble | 0.886 | 0.682 | 0.549 |
| Stack-Ensemble ★ | **0.902** | **0.690** | **0.553** |
| *Baselines:* | | *Tabular AutoML + Feature Engineering* | |
| AG-Weighted | 0.697 | 0.510 | 0.154 |
| AG-Stack | 0.700 | 0.513 | 0.155 |
| AG-Weighted+ N-Gram | 0.872 | 0.679 | 0.471 |
| AG-Stack+ N-Gram | 0.877 | 0.688 | 0.519 |
| H2O AutoML | 0.692 | 0.551 | 0.343 |
| H2O AutoML + Word2Vec | 0.843 | 0.627 | 0.445 |
| H2O AutoML + Pre-Embedding | 0.769 | 0.567 | 0.427 |

Table S1: Alternative summary of AutoML results over our multimodal benchmark, where performance on each dataset is separately averaged over the **binary** classification tasks (i.e. average AUC), **multiclass** classification tasks (i.e. average accuracy), and **regression** tasks (i.e. average $R^2$). See Table 3 for additional details.

AutoGluon also automatically infers the type of each feature via simple yet effective heuristics. One decision particular to our multimodal applications is when to designate a column of string values as a categorical vs. text feature. In this work, we simply threshold based on the number of unique values in the column, such that commonly reoccurring strings are treated as discrete categories rather than unstructured text. We choose the threshold to be 20 in all presented experiments, based on visually confirming the inferred feature types with this threshold agree with our intuition regarding which columns should be handled as text.

While the aforementioned steps are used to report the feature types for each dataset listed in Table 2, we note that our benchmark does *not* require systems to treat certain columns as particular data types. Feature type inference is instead left up to individual methods, since automatically identifying the best way to treat certain columns remains an important research question.

## B.2 Network Architectures

In this paper, we used a single-hidden-layer MLP as the basic building block for encoding features and projecting the hidden states. It has one bottleneck layer and uses layer normalization. We use the leaky ReLU activation (with slope set to 0.1) for all basic MLP layers mentioned throughout the paper. For the 6-layer Transformer model in *Fuse-Early*, we used the GeLU activation like Devlin et al. [14]. We set the number of units, heads, and hidden size of FFN (the feedforward layers) in this Transformer to be 64, 4, 256 correspondingly. For the categorical features, we use an encoding network that is similar to the factorized embedding in ALBERT [89], in which we use an embedding layer with 32 units and then project it with a basic MLP layer that has 64 bottleneck units. We further set the number of output units in the basic MLP to be the same as the token-embeddings used in the pretrained Transformer model (i.e., ELECTRA or RoBERTa) so that all vectors belong to the same space.

In the *Fuse-Late* variant, we further concatenate all encoded categorical features and encode them with a second basic MLP layer. Numeric features are concatenated and encoded with one basic MLP layer. These MLP layers all utilize 128 bottleneck units and their output unit number matches the dimensionality of token embeddings for the pretrained Transformer. The total number of parameters for the *All-Text*, *Fuse-Late*, and *Fuse-Early* multimodal network variants are: 109.0 million, 109.1 million, and 109.3 million correspondingly. Thus, these three model variants have comparable costs.

### B.3 Neural Network Optimization

All text/multimodal neural networks are trained with the slanted triangular learning rate scheduler [84] with initial learning rate set to 0.0, the maximal learning rate set to $5 \times 10^{-5}$ and warmup set to 0.1. We use a batch size of 128, $10^{-4}$ weight decay, and the AdamW optimizer. Text/multimodal networks are trained for 10 epochs and we early stop based on their validation performance. These learning rate and weight decay values were determined via grid search on a single smaller (subsampled) dataset that we used for early initial experiments.

### B.4 Details of AutoGluon Tabular Models in the Stack Ensemble

For better efficiency, we considered just the following tabular models when running AutoGluon [16]:

- Fully-connected Neural Network (MLP) with ReLU activations [16].

- LightGBM model with default hyperparameters (GBM) [38].

- A second LightGBM model with a different set of hyperparameter values. By default, AutoGluon uses this second model in conjunction with the first LightGBM model.

- An implementation of Extremely Randomized Trees from the LightGBM library [23].

- CatBoost gradient boosted trees for sophisticated handling of categorical features [47].

To avoid overfitting in stacking, all models are trained with 5 fold cross-validation (bagging) as described by Erickson et al. [16]. For classification tasks, the outputs of each base model which are aggregated in the ensemble are taken to be predicted class probabilities.

### B.5 Notes on Hyperparameter Tuning

Note that hyperparameter tuning was not a major focus in the preliminary study conducted in this paper. Standard hyperparameter tuning strategies [97] are readily applicable to our multimodal setting, and the experiments presented here could easily employ the advanced Bayesian optimization techniques available in AutoGluon [99]. We expect the performance of all of our proposed AutoML strategies will grow even better with time devoted to hyperparameter tuning. However in this paper we did not conduct such a search and simply used the default hyperparameters supplied by AutoGluon for tabular models, which are already highly performant [16], and the text/multimodal network hyperparameters are listed here and are viewable in our released code. Over just a few datasets, we found that relative performance of different strategies did not qualitatively differ with other reasonable manually-chosen hyperparameter settings (i.e. hyperparameter values known to generally work well for these specific models such as alternative popular learning rate schedules or small changes to the size of the networks).

Rather than only reporting a couple thoroughly-tuned results, we instead preferred to spend our time/compute budget to explore more modeling strategies over more datasets. Note that all H2O AutoML variants reported in Table 3 relied on extensive hyperparameter sweeps (automatically used within H2O), and yet were still unable to outperform some of the other untuned methods we considered. This further supports the claim that our benchmark has helped us identify a broadly performant strategy for multimodal AutoML.

### B.6 Compute Details

All experiments were run on Amazon Web Services EC2 cloud instances (P3.2xlarge). Each instance has two NVIDIA V100 Tensor Core GPUs. About 2000 hours of total compute was required for all experiments presented in this paper (18 instances used for about a week). Given a limited compute budget, we believe more meaningful conclusions may be drawn by running more algorithms over more datasets rather than replicate runs of different seeds/splits on just a few (less diverse) datasets. We also did not include any small datasets in our benchmark for which replicate runs would otherwise be required to get statistically stable results.

## C  Feature Importance Analysis

Feature importance can help us understand what drives a ML system's accuracy and whether text fields in a dataset are worth their overhead. For two representative datasets from our benchmark, we compute *permutation feature importance* [7] for our trained models, which is defined as the drop in prediction accuracy after values of only this feature (which are entire text fields for a text column) are shuffled in the test data (across rows). We only shuffle original column values so our importance scores are not biased by preprocessing/featurization decisions (except in how these directly affect model accuracy).

Figure S4 shows that both our *Multimodal-Net* and *Stack-Ensemble* containing this network may rely more heavily on text features than the *AG-Stack+N-Gram* baseline. With more powerful modeling of text fields, models often begin to rely more heavily on the text fields. An exception here is the *brand_name* feature in mercari, but this feature usually contains just a single word in its fields. Furthermore, the *Multimodal-Net* places less importance on the tabular features, demonstrating how purely neural network approaches are less effective for modeling numeric/categorical data compared to alternative tree-based tabular models. It is thus useful to combine both multimodal-Transformer and tabular models in order to ensure we are most effectively modeling both the text and tabular features.

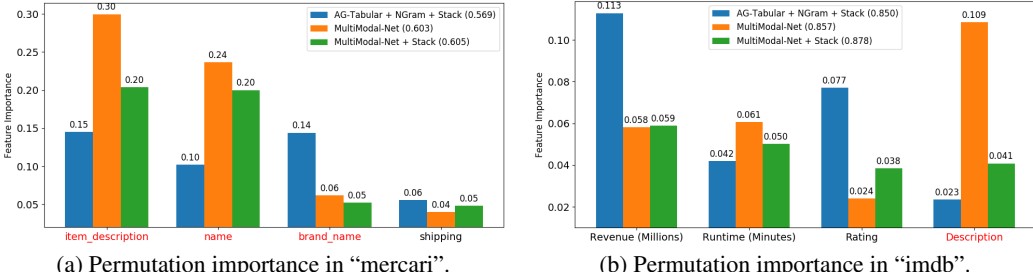

(a) Permutation importance in "mercari".  (b) Permutation importance in "imdb".

Figure S4: Importance of text vs. tabular features for three models in two datasets (text features in red). Here *MultiModal-Net + Stack* corresponds to the *Stack-Ensemble* method from Figure S3c.

## D   Datasheet for our Multimodal Text/Tabular Benchmark

To avoid redundancy, we only provide details here not covered elsewhere in the paper or our benchmark repository. Table 2 lists statistics of each dataset. For details on how each dataset was collected, please refer to the original source linked in our benchmark repository.

**How were datasets selected for the benchmark?** The 18 datasets in our benchmark represent all of the public text/tabular datasets we could find that do not violate our exclusion criteria and satisfy our main desiderata: the dataset must entail a meaningful prediction problem with real enterprise data (as opposed to contrived toy task without real-world application). Note that we only consider tabular datasets that contain text fields, which is a small fraction of publicly available tabular datasets (even though such data are ubiquitous in private enterprises). Our dataset search was conducted over the following sources: Kaggle, MachineHack, UCI ML Repository; the first two are the best sources of publicly available enterprise datasets (with meaningful prediction problems) that we are aware of.

Within each source, we searched for datasets matching the keyword "text" in their meta-data/descriptions for consideration in our benchmark (although the majority such datasets either had no tabular features or failed to provide the original raw text presenting only a featurized version such as bag-of-words). We also conducted some dataset searches via Google, but did not find serious candidates for the benchmark via this avenue. Beyond the primary requirement that data must stem from a real enterprise application with a meaningful classification/regression task, our other exclusion criteria ensured each dataset in the benchmark has: IID examples, non-prohibitive licensing, some text fields beyond just 1-2 words and in the English language (for simplicity), sample size of at least 1000, and predictive signal across both text and tabular (numeric+categorical) modalities (meaning one modality does not appear entirely useless for the prediction problem, evaluated via preliminary *AG-Stack+Ngram* runs without each modality).

**For what purpose were the benchmark datasets created?** We collected the datasets in this benchmark to evaluate supervised machine learning (classification/regression) algorithms designed to jointly operate on text and tabular features. The original versions of these data were also initially created primarily for a similar purpose.

**Who created this benchmark? Who funded its creation?** The authors of this paper, all scientists employed by Amazon, curated this benchmark. Curating the benchmark did not cost significant money, and the benchmark data are currently hosted on cloud servers (S3) provided by Amazon. The original data sources were created/curated/funded by various companies/individuals, please refer to each individual source for more details.

**Do the datasets contain all possible instances or are they a sample (not necessarily random) of instances from a larger set?** Each dataset is a sample of instances from a larger set. We caution these samples may not be at all representative of the larger set, and thus the benchmark should not be used to draw domain-specific conclusions/insights through scientific data analysis of individual datasets.

**Is any information missing from individual instances?** Yes there are many missing fields in certain datasets. It is unclear why they are missing or if the missingness mechanism satisfies the missing at random assumption.

**Are relationships between individual instances made explicit?** For evaluating ML performance, we simply assume the data are IID. However this may be violated by certain datasets. For example, product datasets may contain near duplicate products and products may be related (reviewed by the same users, price of a product can affect price of others, etc.). We do not explicitly know the relationships between instances in these data.

**Are there recommended data splits (e.g., training, development/validation, testing)?** Yes the benchmark provides a recommended training/test split, but ML systems are free to split validation data from the training set as they see fit. The split was done randomly (stratified based on labels for classification) to best reflect an IID setting for which supervised learning methods are primarily

intended.

**Does the benchmark contain data that might be considered confidential?**   Not to our knowledge, but it is possible that a person entered confidential information into the text fields (although they knew these would be publicized).

**Does the benchmark contain data that, if viewed directly, might be offensive, insulting, threatening, or might otherwise cause anxiety?**   The data are mostly non-offensive data used for business purposes. Exceptions are the text fields in the *jigsaw* dataset, which contain toxic online comments, and the *channel*/*pop* datasets, which contain news article titles that may be anxiety-inducing. Furthermore, some of the user reviews of products may be offensive to certain people, although we did not spot any.

**Does the benchmark relate to people?**   Yes some datasets contain information from people. These all stem from commercial sources where people upload their data intentionally to share it with the world (e.g. user reviews, Kickstarter fundraising, public questions, etc.). There is no sensitive/personal information in these data, beyond what a person intended to publicize.

**Is it possible to identify individuals, either directly or indirectly from the benchmark?**   Yes it may be possible as some datasets contain text fields where an individual may have entered arbitrary information (although they knew the information would appear publicly).

**Does the benchmark contain data that might be considered sensitive in any way?**   Not to our knowledge given all this data was already publicly available, but it is possible given the nature of free form text fields.

**How did you process the data from the original sources? Is the software used to preprocess/clean the datasets available?**   We processed each dataset from the original source using the publicly available scripts in the **scripts/data_processing/** folder of our benchmark GitHub repository. To create versions for our benchmark, we omitted certain features (columns), badly formatted or duplicated rows and subsampled overly large datasets.

**Have the benchmark data been used for any tasks already?**   Yes many of the datasets have been used to evaluate ML systems, some through formal prediction competitions. Other datasets have been used to demonstrate data analysis techniques. For the datasets originally stemming from Kaggle, one can find some of the previously considered tasks in the discussion forum or notebooks associated with the original dataset.

**What (other) tasks could the benchmark data be used for? Are there tasks for which these data should not be used?**   We recommend these datasets only be used for evaluation of machine learning algorithms. One could select different target variables in each dataset to create new prediction tasks to evaluate, but these will likely be less practically meaningful (i.e. representative of a real application) than the target variable we have selected for each dataset. Also note that none of the datasets has extremely large sample-size (say over a million), so modeling conclusions drawn based on this benchmark may not translate to applications with massive datasets.

**Will the benchmark be distributed to third parties outside of the entity on behalf of which the dataset was created?**   Yes the benchmark is made publicly available.

**Have any third parties imposed IP-based or other restrictions on the data?**   Yes please refer to the licenses corresponding to each original data source (linked from our repository) for more details.

**Do any export controls or other regulatory restrictions apply to the dataset or to individual instances?**   Not to our knowledge.

**How can the curators of the benchmark be contacted?**   You can open a GitHub issue at the benchmark repository, or email the authors of this paper.

**Will the benchmark be updated (e.g. to correct errors, add new datasets, add/delete instances)?**   Yes updates will be done via GitHub and publicly announced there.

**If others want to extend/augment/build on/contribute to the dataset, is there a mechanism for them to do so?**   Yes anybody may open Pull Request with desired changes on GitHub.

## Additional References for the Appendix

[6] M. Blohm, M. Hanussek, and M. Kintz. Leveraging automated machine learning for text classification: Evaluation of automl tools and comparison with human performance. *arXiv preprint arXiv:2012.03575*, 2020.

[9] R. Caruana, A. Niculescu-Mizil, G. Crew, and A. Ksikes. Ensemble selection from libraries of models. In *International Conference on Machine Learning*, 2004.

[11] K. Clark, M.-T. Luong, Q. V. Le, and C. D. Manning. ELECTRA: Pre-training text encoders as discriminators rather than generators. In *International Conference on Learning Representations*, 2020.

[14] J. Devlin, M.-W. Chang, K. Lee, and K. Toutanova. BERT: Pre-training of deep bidirectional transformers for language understanding. In *NAACL HLT*, 2019.

[15] J. Eisenstein. Natural language processing, 2018.

[16] N. Erickson, J. Mueller, A. Shirkov, H. Zhang, P. Larroy, M. Li, and A. Smola. Autogluon-tabular: Robust and accurate automl for structured data. *arXiv preprint arXiv:2003.06505*, 2020.

[76] R. Fakoor, P. Chaudhari, J. Mueller, and A. J. Smola. Trade: Transformers for density estimation. *arXiv preprint arXiv:2004.02441*, 2020.

[18] R. Fakoor, J. Mueller, N. Erickson, P. Chaudhari, and A. J. Smola. Fast, accurate, and simple models for tabular data via augmented distillation. In *Advances in Neural Information Processing Systems*, 2020.

[78] M. Feurer, A. Klein, K. Eggensperger, J. Springenberg, M. Blum, and F. Hutter. Efficient and robust automated machine learning. In *Advances in Neural Information Processing Systems*, 2015.

[79] M. Feurer, A. Klein, K. Eggensperger, J. T. Springenberg, M. Blum, and F. Hutter. Auto-sklearn: efficient and robust automated machine learning. In *Automated Machine Learning*, pages 113–134. Springer, 2019.

[80] T. Gebru, J. Morgenstern, B. Vecchione, J. W. Vaughan, H. Wallach, H. Daumé III, and K. Crawford. Datasheets for datasets. *arXiv preprint arXiv:1803.09010*, 2018.

[23] P. Geurts, D. Ernst, and L. Wehenkel. Extremely randomized trees. *Machine learning*, 63(1): 3–42, 2006.

[25] C. Guo and F. Berkhahn. Entity embeddings of categorical variables. *arXiv preprint arXiv:1604.06737*, 2016.

[28] H2O.ai. NLP with H2O. `https://docs.h2o.ai/h2o-tutorials/latest-stable/h2o-world-2017/nlp/index.html`.

[84] J. Howard and S. Ruder. Universal language model fine-tuning for text classification. In *Proceedings of the 56th Annual Meeting of the Association for Computational Linguistics*, 2018.

[33] X. Huang, A. Khetan, M. Cvitkovic, and Z. Karnin. Tabtransformer: Tabular data modeling using contextual embeddings. *arXiv preprint arXiv:2012.06678*, 2020.

[36] H. Jin, Q. Song, and X. Hu. Auto-keras: An efficient neural architecture search system. In *KDD*, 2019.

[38] G. Ke, Q. Meng, T. Finley, T. Wang, W. Chen, W. Ma, Q. Ye, and T.-Y. Liu. LightGBM: A highly efficient gradient boosting decision tree. In *Advances in Neural Information Processing Systems*, 2017.

[39] G. Ke, Z. Xu, J. Zhang, J. Bian, and T.-Y. Liu. Deepgbm: A deep learning framework distilled by gbdt for online prediction tasks. In *KDD*, 2019.

[89] Z. Lan, M. Chen, S. Goodman, K. Gimpel, P. Sharma, and R. Soricut. ALBERT: A lite bert for self-supervised learning of language representations. In *International Conference on Learning Representations*, 2020.

[42] E. LeDell and S. Poirier. H2o automl: Scalable automatic machine learning. In *ICML Workshop on Automated Machine Learning*, 2020.

[44] Y. Liu, M. Ott, N. Goyal, J. Du, M. Joshi, D. Chen, O. Levy, M. Lewis, L. Zettlemoyer, and V. Stoyanov. RoBERTa: A robustly optimized bert pretraining approach. *arXiv preprint arXiv:1907.11692*, 2019.

[45] K. Lopuhin and P. Jankiewicz. 1st place solution to mercari price suggestion challenge. `https://github.com/pjankiewicz/mercari-solution`, 2018.

[93] T. Mikolov, K. Chen, G. Corrado, and J. Dean. Efficient estimation of word representations in vector space. *arXiv preprint arXiv:1301.3781*, 2013.

[47] L. Prokhorenkova, G. Gusev, A. Vorobev, A. V. Dorogush, and A. Gulin. CatBoost: unbiased boosting with categorical features. In *Advances in Neural Information Processing Systems*, 2018.

[50] C. Raffel, N. Shazeer, A. Roberts, K. Lee, S. Narang, M. Matena, Y. Zhou, W. Li, and P. J. Liu. Exploring the limits of transfer learning with a unified text-to-text transformer. *Journal of Machine Learning Research*, 21:1–67, 2020.

[51] J. Rodriguez. pytorch-widedeep, deep learning for tabular data i: data preprocessing, model components and basic use. `https://jrzaurin.github.io/infinitoml/2020/12/06/pytorch-widedeep.html`, 2020.

[97] B. Shahriari, K. Swersky, Z. Wang, R. P. Adams, and N. De Freitas. Taking the human out of the loop: A review of bayesian optimization. *Proceedings of the IEEE*, 104(1):148–175, 2015.

[52] Y. Shan, T. R. Hoens, J. Jiao, H. Wang, D. Yu, and J. Mao. Deep crossing: Web-scale modeling without manually crafted combinatorial features. In *Proceedings of the 22nd ACM SIGKDD international conference on knowledge discovery and data mining*, 2016.

[99] L. C. Tiao, A. Klein, C. Archambeau, and M. Seeger. Model-based asynchronous hyperparameter optimization. *arXiv preprint arXiv:2003.10865*, 2020.

[58] M. J. Van der Laan, E. C. Polley, and A. E. Hubbard. Super learner. *Statistical applications in genetics and molecular biology*, 6(1), 2007.

[60] A. Vaswani, N. Shazeer, N. Parmar, J. Uszkoreit, L. Jones, A. N. Gomez, L. Kaiser, and I. Polosukhin. Attention is all you need. In *Advances in Neural Information Processing Systems*, 2017.

[62] A. Wan, L. Dunlap, D. Ho, J. Yin, S. Lee, H. Jin, S. Petryk, S. A. Bargal, and J. E. Gonzalez. NBDT: Neural-backed decision tree. In *International Conference on Learning Representations*, 2021.

[103] D. H. Wolpert. Stacked generalization. *Neural networks*, 5(2):241–259, 1992.