# OpenReview forum: "Benchmarking Multimodal AutoML for Tabular Data with Text Fields"
_NeurIPS.cc/2021/Track/Datasets_and_Benchmarks/Round2 — NeurIPS 2021 Datasets and Benchmarks Track (Round 2)_

### Official Review · Reviewer_EqSr · 2021-09-17
**Paper introduces a valuable benchmark**

**Rating:** 6
**Confidence:** 2

**Strengths:**

It seems valuable to evaluate, hence motivate, AutoML strategies for their capability to handle features from multiple modalities. The experiments demonstrate the incremental predictive performance once models are tailored towards leveraging textual features.
The discussion of the importance/impact of this in practice is not made very concrete in the introduction. Adding this discussion would strengthen the paper.

**Weaknesses:**

Little analyses are presented that give insight into the datasets besides the number of features per data type. A better understanding of these datasets would be informative for analysing future methods benchmarked on these datasets.

The choice for metrics per dataset is weakly motivated; the standard metrics per prediction task class are chosen, disregarding for example class imbalance per-dataset. A few more comments on this in “correctness”.

Although the argument for AutoML benchmarks with a variety of data types is convincing, the motivation in the paper could be strengthened.  For example, an aggregate of feature importances across datasets, e.g. based on the analyses in Appendix C, could support the motivation. Alternatively, a reference to Appendix C in the main paper would be good.

**Additional Feedback:**

The descriptions of the datasets seem inconsistent. All descriptions cover the prediction task, but some include more information of the dataset source and what they are used for. Are the other datasets from Kaggle not used in competitions?

**Clarity:**

The paper, in general, reads well it is hard to make comparisons across datasets and methods based on Table 3. Should this table be exhaustive, or provide easy comparison and key takeaways of evaluations of a few methods? The latter seems more interesting to the reader in this paper. A visual representation of the evaluation would improve this, although some methods might need to be suppressed.

**Correctness:**

The methods benchmarked in this paper seem implemented correctly. I would expect that important preprocessing steps as implemented in the benchmark datasets would instead be part of an AutoML pipeline assessed with this benchmark. For example, it is noted that “we simply rescale and center numeric features and impute their missing values via their average.” But it makes more sense if these are left as a challenge for AutoML pipelines, e.g. imputing missing values with a feature’s average might not necessarily be the best strategy for all datasets. Similar for removing “non-predictive features”, duplicated rows, and highly correlated features, as mentioned in the main paper. I would expect pipelines to solve these tasks automatically.

The evaluation metrics correspond to “standard” metrics for these modeling tasks. I have a few remarks about this choice:
- Some of the datasets are originally accompanied with other metrics (e.g. *house* and *qaa*), it is not explained why other choices are made for these datasets.
- For multiclass classification tasks accuracy might not be accurate, especially in cases of class imbalance. In these cases, I believe more common metrics are precision, recall, F1-scores, etc.
- R2 for regression seems fine to use although it is not bounded by [0,1] as stated.


**Documentation:**

The benchmark provides links to the source datasets and processing scripts to reproduce the processed datasets and train/test splits. Although the seeds for making the splits are not provided in these scripts, these procedures should result in the same datasets used for the benchmark, only the train/test splits should be reconstructed.

The paper in its current form does not list the sources of the dataset, but instead embeds links to the original websites. It seems better to add a column with the data source so that readers do not have to follow 18 links to find this.

It is mentioned that the benchmark will be persisted by GitHub, but it seems to depend on either the existence of the original source files and processing code or on the AWS S3 bucket. It is not mentioned how persistence of this S3 bucket is guaranteed.


**Ethics:**

The licenses of the datasets used and distributed in this benchmark are considered the responsibility of the user, which limits usage. It is also not clear for some of the datasets if their licenses allow distribution. For example, the kick dataset was provided with the explicit “Other” license while the MachineHack datasets seem locked behind registration. Given that the datasets are stored and retrieved from an S3 bucket it seems good to check if this is allowed. An alternative could be to just link to the original data sources, and ensure that the processing scripts are efficient, easily usable, understandable, and yield consistent results (e.g. with split seeds).

**Relation To Prior Work:**

Sufficient detail is provided to understand the added value of the benchmark. Prior benchmarks for AutoML incorporate only categorical and/or numerical variables. The cornerstone of this benchmark is that textual variables exhibit significant signals towards prediction tasks. A few such datasets exist but are not considered fit for purpose due to their size or representativeness of “modern applications”.

**Summary And Contributions:**

This paper describes a benchmark focused on assessing AutoML strategies on datasets with features of multiple modalities, i.e. numerical, categorical and textual. The benchmark consists of 18 datasets originally from platforms like Kaggle, cover classification and regression tasks, and match each task to a standard metric. The evaluation of several multimodal AutoML methods show that extracting signal from textual features improves over numerical-only, which confirms the importance of this benchmark.

----

I thank the authors for their response and appreciate the revised paper. I have updated my rating accordingly.

---

> ### Author Response · Authors · 2021-09-30
> **Response to Reviewer EqSr**
>
> > Paper introduces a valuable benchmark
>
> We are happy to hear the reviewer finds our benchmark valuable, and have added  more in-depth explanations of each dataset, motivations for text/tabular modeling, and justification for the evaluation metrics (see below). As these seem to be the main concerns of your review, we hope you will consider raising your score based on these and other improvements made in the revised paper.
>
> > A better understanding of these datasets would be informative for analysing future methods benchmarked on these datasets. The descriptions of the datasets seem inconsistent. All descriptions cover the prediction task, but some include more information of the dataset source and what they are used for. Are the other datasets from Kaggle not used in competitions?
>
> Not all datasets in our benchmark come from prediction competitions, we merely happened to source many datasets from competitions because they are the best source of public text/tabular data from real commercial applications.
>
> Our revision has added significantly expanded descriptions of the datasets in Section 3.1, including the dataset source, why the prediction problem is meaningful for real applications, and other interesting properties (as much as page limits will allow). Where applicable, we have also cited the original paper introducing the dataset for interested readers to reference.
>
>
> > The choice for metrics per dataset is weakly motivated; the standard metrics per prediction task class are chosen, disregarding for example class imbalance per-dataset. A few more comments on this in “correctness”. The evaluation metrics correspond to “standard” metrics for these modeling tasks. I have a few remarks about this choice:
> > - Some of the datasets are originally accompanied with other metrics (e.g. house and qaa), it is not explained why other choices are made for these datasets.
> > - For multiclass classification tasks accuracy might not be accurate, especially in cases of class imbalance. In these cases, I believe more common metrics are precision, recall, F1-scores, etc.
> > - R2 for regression seems fine to use although it is not bounded by [0,1] as stated.
>
> We believe that using accuracy to evaluate the performance of multiclass classification tasks is a proper choice (not necessarily the only suitable choice of course). In fact, for multiclass classification, **micro F1 score is equivalent to accuracy**. Accuracy is also used in many other well-established benchmarks that involve multiclass classification like ImageNet and many of the tasks in GLUE/SuperGLUE.
>
> For binary classification, we report ROC-AUC rather than accuracy to alleviate potential data imbalance problems. ROC-AUC is one of the most popular metrics used in binary classification and has been adopted in many prediction competitions as a metric that is hard to game and appropriately reflects performance in many real applications.
>
> Note that the "house" dataset is originally accompanied with the Root Mean Squared Logarithmic Error (RMSLE) in Kaggle. For the version in our benchmark, we preprocessed the dataset by taking logarithm of the house prices so that comparing the r2 score of the log-price is monotonically related to the RMSLE used in Kaggle (either metric will result in the same ranking of methods). Our revision clarifies that r2 lies in $[0,1]$ for reasonable predictions, thanks for pointing out that error.
>
> Overall the choice of metrics for such a benchmark is very important, but likely no single optimal choice exists. For example, Reviewer pDGS finds our choice of metrics appropriate, highlighting the subjective nature of evaluation metrics in a benchmark. Thus we have opted to use the standard metrics most commonly used for each type of prediction problem. We prefer that our benchmark uses standard metrics for ease of interpretation and simplifying future research (otherwise researchers may need to spend time adding special tricks per dataset to adapt their models for targeting specific metrics, e.g.   label-smoothing commonly improves log-loss). To ensure ease of interpretation/usage, our benchmark therefore has replaced the metric for a few datasets stemming from competitions that originally used less standard metrics (like qaa which used Spearman correlation).

---

> > ### Author Response · Authors · 2021-09-30
> > **Response to Reviewer EqSr cont.**
> >
> > > For example, it is noted that “we simply rescale and center numeric features and impute their missing values via their average.” But it makes more sense if these are left as a challenge for AutoML pipelines, e.g. imputing missing values with a feature’s average might not necessarily be the best strategy for all datasets.
> >
> > Good point. We have updated paper to clarify that:
> > The tabular MLP networks in AutoGluon-Tabular (the AutoML solution around which our experiments are based) also only use this simple preprocessing. We thus also utilized the same preprocessing for other neural networks evaluated in our experiments for controlled comparison. Note this was only done for the experiments presented here; the actual datasets in our benchmark have not been preprocessed in this manner, and the benchmark leaves preprocessing as a challenge for future AutoML systems to address as they see fit.
> >
> > We have also updated the Discussion to include statement:
> > "Future research should investigate various data preprocessing pipelines, which are known to play an important role in AutoML".
> >
> >
> > > The experiments demonstrate the incremental predictive performance once models are tailored towards leveraging textual features. The discussion of the importance/impact of this in practice is not made very concrete in the introduction. Adding this discussion would strengthen the paper.
> >
> > As suggested, our revision has added a motivating statement:
> >
> > Table 3 shows the accuracy for some datasets significantly improves when models utilize the text features rather than ignoring them. Predictive performance of AG-stack (baseline tabular model that ignores text) vs. Stack-Ensemble (our extension that leverages text) is 0.098 vs. 0.605 on mercari, 0.670 vs. 0.797 on kick, and 0.175 vs. 0.836 on wine. On these datasets, modeling the tabular features brings clear improvements over the text alone given the performance of Text-Net (our best text Transformer model that ignores tabular features) is only: 0.581 on mercari, 0.766 on kick, and 0.807 on wine. In certain applications, accuracy improvements of this magnitude may have significant commercial value, thus highlighting the benefits of multimodal modeling of text/tabular data.
> >
> >
> > > It is mentioned that the benchmark will be persisted by GitHub, but it seems to depend on either the existence of the original source files and processing code or on the AWS S3 bucket. It is not mentioned how persistence of this S3 bucket is guaranteed.
> >
> > The AWS S3 bucket containing the raw data will be persisted via resources provided by AWS. Even if this fails for some reason down the line (as is theoretically possible with most academic data sources), the scripts separately persisted in the GitHub suffice for the benchmark to be recreated from the linked original data sources. As a 3rd backup layer, multiple copies of the original versions of these datasets are also already hosted in various places online.
> >
> >
> > > The licenses of the datasets used and distributed in this benchmark are considered the responsibility of the user, which limits usage. It is also not clear for some of the datasets if their licenses allow distribution. For example, the kick dataset was provided with the explicit “Other” license while the MachineHack datasets seem locked behind registration. Given that the datasets are stored and retrieved from an S3 bucket it seems good to check if this is allowed.
> >
> > We are unable to remove the restriction that "The licenses of the datasets used and distributed in this benchmark are considered the responsibility of the user". That said, this should not limit the use of these datasets for ML research. We have indeed checked that distribution is allowed for these datasets, and have provided proper attribution where required by the dataset creator. In fact, multiple copies of the original versions of these datasets also already exist in various places online.

---

### Official Review · Reviewer_Ag1g · 2021-09-18

**Rating:** 6
**Confidence:** 3

**Strengths:**

- Collection of diverse datasets for three different tasks
- Comprehensive evaluation of several tabular/text-based models and their combinations
- Results highlight the heterogeneous nature of the benchmark

**Weaknesses:**

- While the datasets themselves are quite diverse, I think the evaluation's methodology does not consider an important aspect: training a model with data from each dataset in the benchmark and showing that it performs well still only tells us that the model performs well on domains it has already seen. It is intractable to cover all the domains and applications  (and as the authors note, their datasets are largely related to business applications), so it might be worth considering an evaluation where some datasets (of already trained tasks) remain completely hidden. To me, this would be a more accurate evaluation of an AutoML model -- as I normally do not have any training data for the application I want to use my model for.
- As different metrics have been employed for different tasks,  averaging the scores (as in Table 3) over all datasets does not seem sound
- It would be nice to know whether the format of the tables has been restricted in some datasets, e.g. that headers are always only the first row of a table.
- Some peculiarities of selected datasets (such as the extraordinary low scores on the pop benchmark) were not addressed.

**Additional Feedback:**

- It appears all citations after page 1 are somewhat broken as clicking them sends me back to page 1. The issue was reproduced in two different pdf readers.

**Clarity:**

Clearly written, although some points in the experiments section should rather belong into the discussion (e.g. l.340 -l.344). A conclusion is also missing and must be added in the case of a camera-ready version.

**Correctness:**

See above. Experimental design is sound and described in detail in supp. material.

**Documentation:**

Code and dataset are described in detail in their repository.

**Ethics:**

NA.

**Relation To Prior Work:**

- It is mentioned in l.76-77 that recent work on transformers for understanding structured text tables, such as TaBert or TAPAS addresses a different task than the one in the paper, it is not explained in what way this line of research is different. Is it that these models have been largely studied in the context of question answering? It would be good if the authors could explain their argument, as to me these models seem like a very natural candidate to explore in the proposed benchmark.
- It would be good if the authors could cite relevant papers for each dataset they are using, or directly refer to the dataset page. This information can be found in the repository but seems appropriate to be mentioned in the paper itself.

**Summary And Contributions:**

The authors present a novel benchmark of 18 tabular classification/regression datasets, consisting of tables with numerical, categorical, and at least one text column. All datasets are publicly available and have been aggregated from different sources, mainly Kaggle competitions, covering different domains, sample sizes, and feature numbers with the goal to provide a robust benchmark for real application deployment of AI methods -- evaluating a single model on the entire collection of datasets without manual adjustments. The authors evaluate various baselines and more sophisticated model aggregations on their benchmark, highlighting the heterogeneity of their dataset. Their best performing model, a stack ensemble of tabular models and multimodal transformers, also indicates the complementary nature of both methods.

---

> ### Author Response · Authors · 2021-09-30
> **Response to Reviewer Ag1g**
>
> > While the datasets themselves are quite diverse, I think the evaluation's methodology does not consider an important aspect: training a model with data from each dataset in the benchmark and showing that it performs well still only tells us that the model performs well on domains it has already seen. It is intractable to cover all the domains and applications (and as the authors note, their datasets are largely related to business applications), so it might be worth considering an evaluation where some datasets (of already trained tasks) remain completely hidden. To me, this would be a more accurate evaluation of an AutoML model -- as I normally do not have any training data for the application I want to use my model for.
>
> We clarify that each "task" (i.e. dataset) in benchmark is actually a completely different prediction problem that is unrelated to all the other prediction problems in the benchmark. Thus a model trained for one task is not really appropriate to use for any other task. Therefore we only consider the most basic AutoML setting (standard supervised learning) with a provided training set and held-out test set that are IID for each dataset being evaluated. While it might seem that multiple tasks in the benchmark all involve predicting price of some product, these are actually completely different domains and each dataset has completely different features (such that a model trained on one dataset cannot be easily applied to another dataset).
>
>
> > As different metrics have been employed for different tasks, averaging the scores (as in Table 3) over all datasets does not seem sound.
>
> Note we also report mean reciprocal rank. The reader is free to draw conclusions based on this aggregate metric instead. We feel no aggregate metric will be perfect, but wanted to include our average scores in order to highlight *how much* improvement (on average) one method might offer over another. The famous GLUE/SuperGLUE text modeling benchmarks also use these  techniques to report summary metrics.
>
> The popular (OpenML) tabular AutoML benchmark  (https://www.automl.org/wp-content/uploads/2019/06/automlws2019_Paper45.pdf) reports only per-dataset metrics, which we have also provided in our Table 3 (a reader can simply ignore our additional summary metrics if they prefer).
>
> Finally, we have added Table S1 to the supplement in which we report separate aggregate metrics over binary, multiclass, and regression tasks (so aggregation is always with respect to a single metric). Some readers may find these finer-grained summary metrics more informative than the global aggregation over all tasks or the per-dataset metrics. Note that overall conclusions remain the same based on these per-problem-type  averaged metrics.

---

> > ### Author Response · Authors · 2021-09-30
> > **Response to Reviewer Ag1g cont.**
> >
> > > It is mentioned in l.76-77 that recent work on transformers for understanding structured text tables, such as TaBert or TAPAS addresses a different task than the one in the paper, it is not explained in what way this line of research is different. Is it that these models have been largely studied in the context of question answering? It would be good if the authors could explain their argument, as to me these models seem like a very natural candidate to explore in the proposed benchmark.
> >
> > Our revision clarifies this statement with more detail:
> > While seemingly relevant, recent work on Transformers for understanding structured text tables (Yin et al., 2020; Deng et al., 2020) addresses different tasks than the standard classification/regression tasks with text/tabular features we study in this paper (their research in *table understanding* instead treats entire tables as training examples for tasks like semantic parsing of facts/knowledge (Yin et al., 2020), cell filling, or relation extraction (Deng et al., 2020).
> >
> > These table-understanding models like TaBert are thus designed for entirely different tasks (their notion of "table" is for example a table of facts from a Wikipedia page, whereas ours is a table of IID data as encountered in Statistics class), and they are not appropriate for the classification/regression problems in our benchmark.
> >
> >
> > > It would be good if the authors could cite relevant papers for each dataset they are using, or directly refer to the dataset page. This information can be found in the repository but seems appropriate to be mentioned in the paper itself.
> >
> > Our revision now displays links to the original source for each dataset in the significantly expanded expanded dataset descriptions. Note that many of these datasets are not from academic papers, but we have cited the papers where they are available.
> >
> >
> > > It would be nice to know whether the format of the tables has been restricted in some datasets, e.g. that headers are always only the first row of a table.
> >
> > The datasets are provided as standard CSV/Parquet files that can be easily loaded into Python as a pandas dataframe (with headers always present only in the first row of each table).
> >
> >
> > > Some peculiarities of selected datasets (such as the extraordinary low scores on the pop benchmark) were not addressed.
> >
> > Our revision now includes an expanded description of the 'pop' dataset, which includes the statement:
> > This dataset represents a very difficult prediction problem with only weak signal offered by the observed features. It is fundamentally hard to forecast how popular an article will be based only on its title and crude numerical summary statistics. To be comprehensive, an AutoML  benchmark should contain at least one challenging problem like this.
> >
> >
> > > some points in the experiments section should rather belong into the discussion (e.g. l.340 -l.344).
> >
> > As suggested, our revision has moved the following sentence from these lines out of Experiments and into Discussion:
> > "Given the success of pretrained Transformers across NLP, we are surprised to find both N-Grams and word2vec here provide superior text featurization than Pre-Embedding."
> >
> > For the other sentence from these lines ("The last section of Table 3 shows ... superior overall"), we feel the observations about the results of specific experimental comparisons should be part of the experiments section, but we have also summarized these points again in the Discussion section.
> >
> >
> > > A conclusion is also missing and must be added in the case of a camera-ready version.
> >
> > We have added the Conclusion section after the Discussion.

---

### Official Review · Reviewer_pDGS · 2021-09-19
**Important topic, rich experimentation section, but the benchmark itself lacks some of the qualities of a good benchmark.**

**Rating:** 6
**Confidence:** 3

**Strengths:**

Authors tackle the vital problem of measuring how robust an AutoML algorithm is on a wide variety of tabular tasks. The implications of the achieved baseline results can spur further research ideas. In addition, the benchmark itself has chances of being adopted by the research community as an essential metric. The authors' good choice is to align the datasets with the real-world needs, thus helping to increase the capabilities of using AutoML in industrial applications.


**Weaknesses:**

My main concern with the proposed benchmark is the lack of presented criteria for selecting these particular datasets from all available tabular datasets. The only rationale why these 18 datasets should be considered together is presented using a very vague statement regarding their diversity and real-world relatedness. Still, I would expect that the authors performed a broader search, that baseline algorithms with multiple modalities were evaluated, and that based on the explicitly formulated criteria (e.g., dataset size, scores improvement of multimodal models) some of them were selected, and others were rejected. The lack of assuring the stated qualities during the selection process weakens the overall benchmark's impact. **Why should researchers use the provided aggregation of datasets instead of choosing a different subset of available datasets?**

**Another point worth questioning is whether some of the datasets (e.g., ae, jigsaw, fake, house) has enough room for improvement as the baseline model achieves up to 98% accuracy on some of them.** But, again, this seems related to the previous point of explicitly stating qualities of considered datasets, and the datasets featured in the benchmark should remain challenging to measure the field's progress.

Additionally, the `prod, IMDb, channel, house` datasets seem to not provide reliable information for measuring improvements of a multimodal model. After adding the textual information, the score improvement is not significant for AG models. **So, how will these four datasets allow us to understand the importance of multimodal reasoning?**

Another concern I have regarding the experimentation and comparison of different models - **Is the comparison between fuse-early and fuse-late models fair considering parameter count and the number of operations performed?** From what I see in the appendix, the fuse-early method adds more tokens for tabular data, whereas fuse-late approaches allocate more representation capacity to the multimodal information. If the two methods differ significantly in the costs (i.e., if the bigger model performed better), then some of the statements presented in the discussion section may not hold. However, clarifying that they are comparable because, e.g., they feature the same amount of parameters will clear my concerns.

Overall, the paper's contribution to the "benchmarking" side of AutoML models is not rich enough or not described. It seems like the authors gathered some datasets (diversity appears to be a natural side effect and not a planned feature) and performed extensive evaluations on them to prove that they correctly chose them. However, even some of the results point against it. The problem I see is that the experimentation section of the paper should not be the main contribution of a benchmark paper. There are plenty of low hanging fruits that authors can improve in designing the benchmark itself:
1. provision of human performance (to understand the score at which tasks will be considered solved),
2. description of a dataset search process, e.g., where did authors look for the datasets and what phrases/keywords were used? (to understand that this benchmark cover the most important ones)
3. selection process and elimination rules (to understand criteria for choosing the datasets)

**TL;DR**
Despite this, the paper seems to be a solid submission for a top conference, but some of the design choices in the benchmark itself lack rationale. Most importantly – why should one follow your preferences of the 18 datasets presented here? Additionally, how important is it to measure the improvement on already solved tasks?



**Additional Feedback:**

This question does not impact the score, but addressing it may be considered helpful for further reading:
1. The second point in the discussion seems to be highly related to the third point, that in general, stacking models later helps rather than fusing low-level features. Do you have any rationale or intuition behind the lower performance or any suggestions regarding what can be improved for the `Fuse-Early` type of approaches?


**Clarity:**

The paper is extremely well written, a paragraph-to-paragraph transition is very nice, and reading many multi-sentence passages is a pleasure.

**Correctness:**

The experiment seems sound and solid, with only one minor concern about the fairness of comparison. The author presented and described a wide variety of models, and the choices made there are clear. In the benchmark itself, there are some critical issues raised above, but also many design choices are correct (e.g., metrics, the mix of classification vs. regression tasks, datasets from real applications)

**Documentation:**

Yes, the maintenance aspect is cleared in the paper. The Github code looks reproducible, and README is providing an excellent introduction to the repository.

**Ethics:**

I do not see any potential ethical issues.

**Relation To Prior Work:**

Yes, the authors explicitly compare to previous contributions in section 2.

**Summary And Contributions:**

The authors present a benchmark for measuring the progress of AutoML algorithms on already-existing datasets with multimodal features, such as text and categorical/numerical ones. The authors systematically evaluate the single- and multimodal models, provide a wide variety of baselines and demonstrate empirically that understanding both modalities is crucial for achieving good performance.

---

> ### Author Response · Authors · 2021-09-30
> **Response to Reviewer pDGS**
>
> > The authors' good choice is to align the datasets with the real-world needs, thus helping to increase the capabilities of using AutoML in industrial applications. The experiment seems sound and solid, with only one minor concern about the fairness of comparison. The author presented and described a wide variety of models, and the choices made there are clear. In the benchmark itself, there are some critical issues raised above, but also many design choices are correct (e.g., metrics, the mix of classification vs. regression tasks, datasets from real applications). TL;DR Despite this, the paper seems to be a solid submission for a top conference, but some of the design choices in the benchmark itself lack rationale. Most importantly – why should one follow your preferences of the 18 datasets presented here? Additionally, how important is it to measure the improvement on already solved tasks?
>
> We are happy to hear the reviewer finds our paper to be a solid submission overall, and have added far more in-depth explanations to justify the choices made in creating our benchmark (described below). As these seem to be the main concerns of your review, we hope you will consider raising your score based on our justifications and other improvements made in the revised paper.

---

> > ### Author Response · Authors · 2021-09-30
> > **Response to Reviewer pDGS cont.**
> >
> > > My main concern with the proposed benchmark is the lack of presented criteria for selecting these particular datasets from all available tabular datasets.  Why should researchers use the provided aggregation of datasets instead of choosing a different subset of available datasets?
> >
> > In new "How were datasets selected for the benchmark?" section added to Appendix D, our revision clarifies that:
> > The 18 datasets in our benchmark represent all of the public text/tabular datasets we could find that do not violate our exclusion criteria and satisfy our main desiderata: the dataset must entail a meaningful prediction problem with real enterprise data (as opposed to contrived toy task without real-world application). Note that we only consider tabular datasets that contain text fields, which is a small fraction of publicly available tabular datasets (even though such data are ubiquitous in private enterprises). Our dataset search was conducted over the following sources: Kaggle, MachineHack, UCI ML Repository; the first two are the best sources of publicly available enterprise datasets (with meaningful prediction problems) that we are aware of.
> >
> > We are happy to add additional datasets to the benchmark in our revision if the reviewer knows of any other public text/tabular datasets that fit our criteria?
> >
> > We believe our benchmark is valuable precisely because such datasets are not common in public domain (even though they are ubiquitous in enterprise ML applications). Assembling the first comprehensive text/tabular benchmark allowed us to discover the suboptimality of popular techniques like the Pre-Embedding method used by Blohm et al. (2020) and many others. Most important is simply having any public benchmark in this space to keep practitioners from using poor methods like: Pre-Embedding. Our work provides the first text/tabular benchmark (not necessarily the last), which we feel is urgently needed by the community as research in text/tabular modeling has clearly been held back by the lack of such a benchmark. Our revision additionally emphasizes: Further evidence that real problems are well-represented in our benchmark comes from the finding that the top methods revealed over our benchmark are able to rank so highly in various historical text/tabular prediction competitions.
> >
> > Even the popular (OpenML) tabular AutoML benchmark of Gijsbers et al. (https://openml.github.io/automlbenchmark/benchmark_datasets.html) originally only contained 39 basic tabular datasets (out of a much larger universe of possibilities for basic tabular data), as they sought to balance the same key criterion as us ("representative of real-world data science problems") with other criteria like diversity and computational constraints. Benchmark design is nontrivial and even established benchmarks like the tabular AutoML benchmark remain far from perfectly representative today; in contrast, we have simply collected as many public text/tabular datasets (with meaningful prediction problems) as we could find. For any benchmark, there will always be extra datasets that could have been included, but we do not believe their existence means the benchmark is not valuable. A benchmark can be good for evaluating AutoML as long as it contains a diversity of: meaningful prediction tasks, sample sizes, and number/types of features.
> >
> > We also briefly highlight opinions from the other reviewers regarding the benchmark datasets:
> >
> > - Reviewer 5wa3: "The benchmark, consisting of 18 datasets seems sufficient for evaluating models in this Tabular+Text domain."
> >
> > - Reviewer Ag1g says our benchmark offers: "Collection of diverse datasets for three different tasks", "Comprehensive evaluation of several tabular/text-based models and their combinations", and "Results highlight the heterogeneous nature of the benchmark".
> >
> > - Reviewer EqSr: "Paper introduces a valuable benchmark".

---

> > > ### Author Response · Authors · 2021-09-30
> > > **Response to Reviewer pDGS cont.**
> > >
> > > > There are plenty of low hanging fruits that authors can improve in designing the benchmark itself:
> > > > - Provision of human performance (to understand the score at which tasks will be considered solved)
> > > > - Description of a dataset search process, e.g., where did authors look for the datasets and what phrases/keywords were used? (to understand that this benchmark cover the most important ones)
> > > > - Selection process and elimination rules (to understand criteria for choosing the datasets)
> > >
> > > In additional "How were datasets selected for the benchmark?" section added to Appendix D, our revision clarifies that:
> > > Our dataset search was conducted over the following sources: Kaggle, MachineHack, UCI ML Repository; the first two are the best sources of publicly available enterprise datasets (with meaningful prediction problems) that we are aware of. Within each source, we searched for datasets matching the keyword "text" in their metadata/descriptions for consideration in our benchmark (although the majority such datasets either had no tabular features or failed to provide the original raw text presenting only a featurized version such as bag-of-words). We also conducted some dataset searches via Google, but did not find serious candidates for the benchmark via this avenue. Beyond the primary requirement that data must stem from a real enterprise application with a meaningful classification/regression task, our other exclusion criteria ensured each dataset in the benchmark has: IID examples, non-prohibitive licensing, some text fields beyond just 1-2 words and in the English language (for simplicity), sample size of at least 1000, and predictive signal across both text and tabular (numeric+categorical) modalities (meaning one modality does not appear entirely useless for the prediction problem, evaluated via preliminary AG-Stack+Ngram runs without each modality).
> > >
> > > Examples of UCI text/tabular datasets that violated these exclusion criteria include:
> > > - microblogPCU Data Set (non IID examples): https://archive.ics.uci.edu/ml/datasets/microblogPCU
> > > - Drug Review Dataset (prohibitive licensing): https://archive.ics.uci.edu/ml/datasets/Drug+Review+Dataset+%28Drugs.com%29
> > > - Northix Data Set (small sample size): https://archive.ics.uci.edu/ml/datasets/Northix
> > >
> > > Note we did not simply aggregate existing datasets to create the benchmark, but also cleaned some of them in important ways (for example: ensuring a well formulated prediction problem in a dataset not originally intended specifically for ML, or removing duplicate examples as it would not be ok to have duplicates in training and test sets).
> > >
> > > Unlike popular NLP/vision benchmarks, the tasks in our benchmark are not expected to be solvable by a human (they are not based on recognition or common-sense reasoning). Thus we do not believe reporting human prediction performance is meaningful as humans likely fall well below the best achievable ML performance. If we misunderstood the reviewer and you actually mean to report performance achieved by human ML expert investing significant time manually modeling each dataset, we also did not report this because the number would vary based on the person/effort-invested as well as what models were used (or even invented at this point in time).

---

> > > > ### Author Response · Authors · 2021-09-30
> > > > **Response to Reviewer pDGS cont.**
> > > >
> > > > > Another point worth questioning is whether some of the datasets (e.g., ae, jigsaw, fake, house) has enough room for improvement as the baseline model achieves up to 98\% accuracy on some of them.
> > > > > Additionally, the prod, IMDb, channel, house datasets seem to not provide reliable information for measuring improvements of a multimodal model. After adding the textual information, the score improvement is not significant for AG models. So, how will these four datasets allow us to understand the importance of multimodal reasoning?
> > > >
> > > > While the addition of simple N-gram featurization of text does not improve the baseline AG models, that does not mean the textual information is useless in these datasets. We can see that the text clearly provides some predictive signal by observing the Transformer-based methods are able to outperform the AG baseline without text. Thus these datasets represent instances of challenging text fields where it is hard to extract text information in a way that boosts predictive accuracy. Including challenging datasets like this is critical to increase diversity of our text/tabular benchmark. Likewise it is important to include easy datasets (to ensure AutoML solutions are not only performant on the challenging datasets), and it is important to include some datasets where the text provides a lot of predictive signal as well as some where it provides very little predictive signal. The benchmark must contain datasets with all of these various properties in order to comprehensively evaluate text/tabular AutoML systems across the wide range of real-world applications they may face.
> > > >
> > > > For the "easy" datasets pointed out by reviewer (ae, jigsaw, fake, house), it remains unclear how much higher the best achievable accuracy (Bayes error) may be. That said, even if the baseline AG + N-gram solution is already close to this value, we believe these datasets serve as important "sanity checks" in the benchmark to ensure a proposed AutoML pipeline is not failing to match what should be easily obtainable performance. For instance we see the popular Pre-Embedding approach (used as a text-AutoML solution by Blohm et al.) performs poorly on jigsaw & fake, which are important flaws to surface. Our benchmark already contains plenty of harder datasets which remain challenging to measure the field's progress, we simply want to ensure this progress does not neglect the sanity checks present in our current comprehensive benchmark. Note the (OpenML) tabular AutoML benchmark also contains a few easy datasets for this reason (e.g. kr-vs-kp, riccardo, nomao, etc.).

---

> > > > > ### Author Response · Authors · 2021-09-30
> > > > > **Response to Reviewer pDGS cont.**
> > > > >
> > > > > > Another concern I have regarding the experimentation and comparison of different models - Is the comparison between fuse-early and fuse-late models fair considering parameter count and the number of operations performed? From what I see in the appendix, the fuse-early method adds more tokens for tabular data, whereas fuse-late approaches allocate more representation capacity to the multimodal information. If the two methods differ significantly in the costs (i.e., if the bigger model performed better), then some of the statements presented in the discussion section may not hold. However, clarifying that they are comparable because, e.g., they feature the same amount of parameters will clear my concerns.
> > > > >
> > > > > Appendix B.2 of our revision now clarifies that:
> > > > > The total number of parameters for the All-Text, Fuse-Late, and Fuse-Early multimodal network variants are: 109.0 million, 109.1 million, and 109.3 million correspondingly. Thus, these three model variants have comparable costs.
> > > > >
> > > > >
> > > > > > The second point in the discussion seems to be highly related to the third point, that in general, stacking models later helps rather than fusing low-level features. Do you have any rationale or intuition behind the lower performance or any suggestions regarding what can be improved for the Fuse-Early type of approaches?
> > > > >
> > > > > Nice observation! We have added a statement in the Discussion:
> > > > > A general observation across our benchmark is that stacking/fusing models later helps more than fusing low-level features. While this finding contrasts with other multimodal ML research (Radford et al., 2021; Singh et al., 2020), we suspect the primary reason is that most of this other multimodal ML has predominantly has focused on particular matching tasks with mostly image+text data (e.g. image captioning). Such tasks require learning to correlate low-level features of one modality (e.g. specific words) with low-level features in the other modality (e.g. specific pixel regions), and thus early-fusion is a natural modeling strategy. However, our benchmark is composed of commercially relevant classification/regression tasks with text+tabular data, in which we typically predict an auxiliary variable based on the text/tabular features (e.g. the price of a product). For such tasks, we suspect correlating low-level features between modalities is unnecessary and it is more critical that models can adequately summarize/extract the relevant information from each modality before considering the interaction between them. Using our benchmark, future work may more formally investigate such hypotheses.

---

> > > > > > ### Comment · Reviewer_pDGS · 2021-10-04
> > > > > > **Note on updated rating**
> > > > > >
> > > > > > Thank you for addressing my main concerns. I updated the rating after reading the rebuttal and briefly reading through the paper.
> > > > > >
> > > > > > The description of the search process I found in the replies is the most important one that caused an increase in my rating.
> > > > > >
> > > > > > The idea of performing sanity checks makes sense to me, but frankly, I would more explicitly separate them from the rest since the concept is quite different. Although I also believe that authors know what they are doing, and I accept their point of view.
> > > > > >
> > > > > > I believe the paper is worth accepting.

---

### Official Review · Reviewer_5wa3 · 2021-09-20

**Rating:** 6
**Confidence:** 3
**Correctness:** Yes.
**Clarity:** Yes.

**Strengths:**

The paper is well written and easy to understand.

The paper evaluates various baselines including simple "hacks" on the proposed benchmark and also proposes a method that works best on this dataset.

The benchmark, consisting of 18 datasets seems sufficient for evaluating models in this Tabular+Text domain.



**Weaknesses:**

What are the performances of the methods without using the text field? I can see performances of using only text and the combination, but am not sure about the performance without text.

Lack of analysis. While the paper shows that using multi-modal embeddings can help, it’s not obvious why or when. Especially since this paper advocates for the usage of text-inputs, it’d be interesting to understand when it helps and when it does not. Also, the fact that “the majority of additional gains produced by exponential decay”, is something that is hard to understand why this is the case for text models.

Lack of clear learnings. The differences between the methods are quite small and there do not seem to be any major insights gained from using this “new” source of inputs. It'd be great if the paper makes the findings (that results from this dataset) clearer and better differentiate it from the methods applied.

**Additional Feedback:**

The paper reads a bit like an advertisement, e.g.
> The data files are currently hosted in AWS Simple Cloud Storage 150 (S3), which is a reliable and highly-available medium.


> modeling is simply done via AutoGluon-Tabular, an easy-to-use and highly accurate open-source tool for automated supervised learning on tabular data

or
> For real-time applications with latency constraints, AutoGluon provides options to accelerate ensemble inference via pruning or distillation [17]
which seems completely unrelated to what the paper is proposing.


**Documentation:**

Yes.

**Ethics:**

No, it's a benchmark.

**Relation To Prior Work:**

Yes, as far as I can tell.

**Summary And Contributions:**

The paper proposes a new benchmark on evaluating AutoML methods on tabular+text data. The paper accumulated 18 datasets from industrial applications that combine numeric/feature fields with free text-form fields and are either regression or classification tasks. As a motivation for this benchmark the paper shows that adding inputs from text-columns helps increase the performance on this benchmark.

---
After the author response, I feel like my concerns have been partically addressed and I remain at the rating 6. The reason for not going above is that I still believe that the paper does not provide enough analysis of the presented results. It would be great if the authors can state this part stronger and expand on it in their final version, as this will motivate readers of this paper to work on and develop tabular+text methods, which is the goal of this benchmark.

---

> ### Author Response · Authors · 2021-09-30
> **Response to Reviewer 5wa3**
>
> > What are the performances of the methods without using the text field?
>
> Our revision clarifies: Baseline ML performance of tabular models without using any text fields can be established via the AG-Weighted/Stack numbers in Table 3  (without N-Gram).
>
>
> > “the majority of additional gains produced by exponential decay”, is something that is hard to understand why this is the case for text models.
>
> One way to explain this phenomena is that different layers of the pretrained text model capture different types of features. The layers that are closer to the input will capture the basic syntactic structure of the text which is more or less universal across datasets. The upper layers will capture high-level semantics that are more specific to a particular dataset/task. With the exponential decay learning rate, the lower layers will be adapted more slowly than the upper layers, thus retaining more of the learned low-level syntactic text properties acquired via pretraining. As shown by the experiments, this improves the performance.
>
>
>
> > The paper reads a bit like an advertisement, e.g.
> "The data files are currently hosted in AWS Simple Cloud Storage 150 (S3), which is a reliable and highly-available medium"
> or
> "modeling is simply done via AutoGluon-Tabular, an easy-to-use and highly accurate open-source tool for automated supervised learning on tabular data"
> or
> "For real-time applications with latency constraints, AutoGluon provides options to accelerate ensemble inference via pruning or distillation", which seems completely unrelated to what the paper is proposing.
>
> These statements are not seeking to advertise these tools but rather aiming to justify why we chose S3 for hosting the datasets and AutoGluon as a baseline for tabular ML. Our revision has clarified the language used to clarify that we are only stating their advantageous qualities as a justification for our choices in this paper (which seems critical to include). The final statement was originally there to highlight that our Weighted/Stack-Ensemble approaches can be accelerated via AutoGluon's existing ensemble-pruning/distillation functionality. Since the reviewer finds it unclear, we have omitted this statement in the revision.

---

### Author Response · Authors · 2021-09-30
**Revision uploaded (with revised supplement file that contains main text + appendix)**

Thanks for the insightful and detailed comments by all reviewers. We have revised our paper according to the feedback and have uploaded an improved version. Major additions include more in-depth: justification of the many choices we had to make and descriptions of each dataset. We have also added a new Table S1 to Appendix, which provides a finer-grained summary of AutoML performance over our benchmark (broken down by prediction problem type) than the aggregate metrics over all datasets provided in Table 3.
A point-by-point response to reviewer questions is provided below, please let us know if any other questions come up!

---

### Decision · Program_Chairs · 2021-10-09

**Decision:**

Accept

**Comment:**

This paper presents a collection of tabular+text datasets and also introduces new methods for handling the combination of tabular+text. This doesn’t provide a perfect focus on the actual new dataset collection, which was the primary issue that reviewers criticized. The authors added much new information about the actual datasets in an extensive rebuttal phase with all reviewers, in the end leaving the reviewers satisfied. I therefore recommend acceptance now. I encourage the authors to add any further documentation and statistics about the datasets they think could be useful.